# General structural features that regulate integrin affinity revealed by atypical αVβ8

Jianchuan Wang [1,2], Yang Su[1,2], Roxana E. Iacob[3], John R. Engen [3] & Timothy A. Springer [1,2]*

Integrin αVβ8, which like αVβ6 functions to activate TGF-βs, is atypical. Its β8 subunit binds to a distinctive cytoskeleton adaptor and does not exhibit large changes in conformation upon binding to ligand. Here, crystal structures, hydrogen-deuterium exchange dynamics, and affinity measurements on mutants are used to compare αVβ8 and αVβ6. Lack of a binding site for one of three βI domain divalent cations and a unique β6-α7 loop conformation in β8 facilitate movements of the α1 and α1' helices at the ligand binding pocket toward the high affinity state, without coupling to β6-α7 loop reshaping and α7-helix pistoning that drive large changes in βI domain-hybrid domain orientation seen in other integrins. Reciprocal swaps between β6 and β8 βI domains increase affinity of αVβ6 and decrease affinity of αVβ8 and define features that regulate affinity of the βI domain and its coupling to the hybrid domain.

[1] Program in Cellular and Molecular Medicine, Boston Children's Hospital, Boston, MA, USA. [2] Department of Biological Chemistry and Molecular Pharmacology, Harvard Medical School, Boston, MA 02115, USA. [3] Department of Chemistry and Chemical Biology, Northeastern University, Boston, MA 02115, USA. *email: springer_lab@crystal.harvard.edu

Integrins comprise a family of αβ heterodimers with diverse functions in cell adhesion, migration, and signaling. The integrin family was seeded with its first two members, integrins lymphocyte function-associated antigen 1 (integrin αLβ2) and macrophage antigen 1 (integrin αMβ2) with discoveries in 1982 that they had identical β-subunits and distinct α-subunits[1] and in 1985 that their α-subunits, αL and αM, had homologous amino acid sequences[2]. The integrin family grew to its current size in mammals of 24 αβ heterodimers with the cloning of the last β-subunit, β8, in 1991[3] and the last α-subunit, α11, in 1999[4]. Integrins are force-resistant and provide traction for cell migration and mechanical stability for tissues. However, two integrins, αVβ6 and αVβ8, appear to have evolved primarily to activate transforming growth factor-β1 (TGF-β1) and TGF-β3 and bind with high affinity to a RGDLXX(L/I) motif in the TGF-β prodomain. These integrins activate TGF-β by releasing the growth factor (GF) from the prodomain, which otherwise surrounds the GF and holds it in a latent form in which it cannot bind TGF-β receptors[5]. αVβ8 is expressed much more highly in the brain than in any other tissues[3] and especially highly on glial cells, where it localizes to synapses with other glia and neurons and within synaptosomes[6]. αVβ8 in the central nervous system is required to activate TGF-β1 complexed with the milieu molecule LRRC33 on microglia cells and for the maintenance of myelin, axons, and neurons in certain regions of the central nervous system and particularly in motor pathways in the brain and spinal cord[7].

αVβ8 differs from αVβ6 and all other integrins in its coupling to the cytoskeleton. Among the 24 integrin heterodimers, 22 bridge extracellular ligands to the actin cytoskeleton by binding through specific sites in integrin β-subunit cytoplasmic domains to the adaptors talin and kindlin (Fig. 1a–c)[8]. Retrograde actin flow transmits tensile force through such integrins when they bind to extracellular ligands. This tensile force, together with ligand binding, stabilizes integrins in the extended-open conformation, which, depending on the integrin, has 700- to 4,000-fold higher affinity for ligand than the extended-closed or bent-closed conformations[9,10]. Higher affinity results from tightening of the ligand-binding site in the integrin β-subunit βI domain at the β1-α1 loop and α1-helix around the metal ion-dependent adhesion site (MIDAS) (Fig. 1b, c)[11,12].

In contrast, the β8-subunit of integrin αVβ8 has a divergent cytoplasmic domain that binds to the Band 4.1 family (Fig. 1d–f)[13]. Although the conformation of intact αVβ8 is unknown, its ectodomain fragment almost exclusively exhibits an extended conformation[14–16] (Fig. 1e), and thus cannot be activated by tensile force. Furthermore, in contrast to other integrins, binding of ligand to αVβ8 fails to induce swing-out of the hybrid domain; that is, the open headpiece conformation (Fig. 1f)[14–16]. Therefore, we

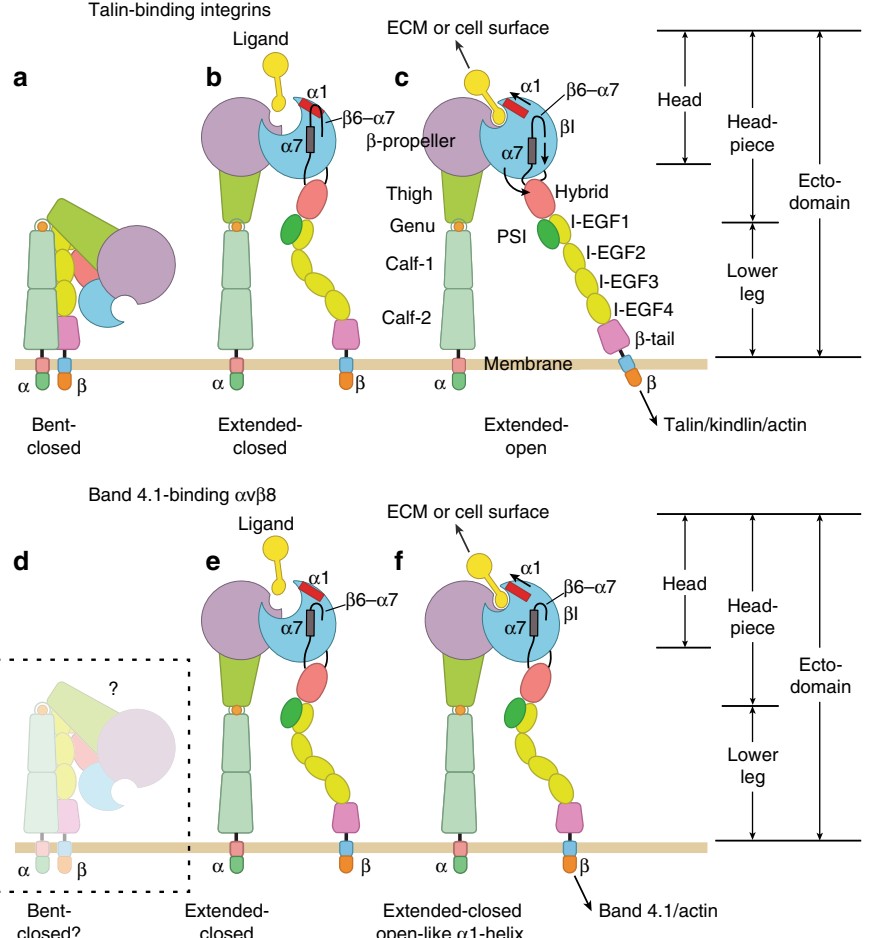

**Fig. 1** Overall integrin conformational states. **a–c** Talin-binding integrins. **d–f** Band 4.1-binding integrin αVβ8. **d** Population of the bent-closed conformation is very low in αVβ8[14–16], whereas this is by far the most populous state on cell surfaces in typical integrins (**a**)[9]. **e** The extended-closed conformation is by far the most populous αVβ8 conformation, at least in solution[14–16]. **f** αVβ8 does not exhibit a ligand-stabilized extended-open conformation like typical integrins (**c**). However, ligand binding induces movement of the α1-helix (and SDL1, which includes the N-terminal portion of the α1-helix and its preceding loop) toward the open state of the βI domain. The β6-α7 loop in the βI domain is unengaged with the α7-helix, enabling opening of the βI domain to occur in the absence of hybrid domain swing-out.

wondered how αVβ8 is able to bind pro-TGF-β1 with an affinity that is unusually high for an integrin[14]. The β8-subunit has Asn instead of Asp at two positions known to coordinate with a $Ca^{2+}$ ion at the adjacent to MIDAS (ADMIDAS) in the βI domain of other integrins; however, both Asn and Asp can coordinate $Ca^{2+}$.

Here, we report structural differences and correlating sequence differences between the βI domains of β8 and talin-activated integrin β-subunits that extend well beyond the Asp and Asn ADMIDAS differences, including the β6-α7 loop, and have not previously been discussed or studied. Mutational exchanges between β8 and β6 βI domains and hydrogen-deuterium exchange (HDX) differences between β8 and β6 suggest that that these sequence motif and structural differences have important roles in affinity regulation and may enable affinity regulation without hybrid domain swing-out in atypical β8 (Fig. 1f). We further find that in typical integrins, the β6-α7 loop has an important role in maintaining the low-affinity state.

## Results

### The structure of integrin αVβ8 and its lack of an ADMIDAS.
An integrin αVβ8 headpiece fragment with high mannose N-glycans was expressed in GnTI-deficient HEK293 cells, purified, crystallized, and soaked with or without the TGF-β1 ligand peptide $G^{213}RRGDLATIHG^{223}$ (Table 1). The β-propeller and thigh domains in αV and the βI and hybrid domains in β8 are resolved in the structure (Fig. 2a, b). The co-crystallized peptide fragment of the TGF-β1 prodomain binds to the interface between the αV β-propeller and β8 βI domain. αVβ8 electron density is poorer in the hybrid domain than in the β-propeller, thigh, and βI domains, and absent in the PSI (plexin, semaphorin, and integrin) and I-EGF$^{-1}$ (integrin-epidermal growth factor-like$^{-1}$) domains, which link to the N- and C-terminal ends of the hybrid domain distal to its interface with the βI domain. In contrast, all β-subunit domains were better defined in αVβ6 headpiece crystal structures (Fig. 2c)[17]. Regions of the β8 hybrid domain that could not be built are missing or dashed in Fig. 2a, b; the β6 hybrid domain in Fig. 2c appears larger because it is entirely built. β8 hybrid domain electron density is better at its interface with the βI than the PSI-I-EGF-1 domains, and variable among independent molecules in asymmetric units (four in unliganded αVβ8 and two in liganded αVβ8).

The β8 βI domain has unique features compared to previously structurally characterized integrin β-subunits, all of which link to the actin cytoskeleton through talin and kindlin, that is, β1, β2, β3, β6, and β7[11,17–19]. The most striking difference is the lack of an ADMIDAS $Ca^{2+}$ ion (Figs. 2a, b and 3). To ensure that lack of an ADMIDAS metal ion was not an artifact related to crystallization of αV integrins at low pH[17], αVβ8 was crystallized at pH 6.7 and $Mg^{2+}$ and $Ca^{2+}$ concentrations were increased during crystal soaking.

For ease of nomenclature here, we define the contiguous sequence of MIDAS and ADMIDAS-coordinating residues in typical integrin β-subunits, DXSXSXXDD (D1-S3-S5-D8-D9), as the β-MIDAS motif, where MIDAS is used in a broad sense to include up to two metal ions (Fig. 3g). In typical integrins, the ADMIDAS metal ion coordinates the sidechains of the two Asp residues (β-MIDAS D8 and D9), the backbone carbonyl of the β-MIDAS S5 residue, and a backbone carbonyl from the β6-α7 loop (Fig. 3c–g). In contrast, β8 has Asn-119 and Asn-120 (N8 and N9) in place of the D8 and D9 Asp residues (Fig. 3a, b, g). Asn carbonyl oxygens can coordinate $Ca^2$, as seen at the SyMBS (synergistic metal ion-binding site) in integrins. However, replacement of β-MIDAS motif D8 and D9 residues with Asn in β8 results in the absence of any negatively charged sidechains to coordinate $Ca^{2+}$ and is likely to be sufficient to explain the lack

**Table 1 αVβ8 headpiece data collection and refinement statistics[a,b].**

| Data collection | Unliganded[a] | Liganded |
|---|---|---|
| Space group | P1 | P2₁ |
| a, b, c (Å) | 144.2, 55.1, 175.1 | 161.2, 53.9, 176.6 |
| α, β, γ (°) | 90.37, 107.0, 90.01 | 90.0, 111.5, 90.0 |
| Unique reflections | 141,394 (10,550) | 71,992 (4828) |
| Redundancy | 1.7 (1.7) | 3.2 (2.2) |
| Resolution (Å) | 50.0-2.66 | 50.0-2.77 |
| | (2.73-2.66) | (2.84-2.77) |
| Completeness (%) | 95.5 (96.8) | 98.5 (89.3) |
| $I/\sigma(I)$ | 5.87 (0.33) | 8.38 (0.61) |
| $R_{merge}$ (%)[c] | 9.4 (192.2) | 11.2 (157.7) |
| $CC_{(1/2)}$ (%)[d] | 99.4 (19.9) | 99.6 (44.1) |
| Wavelength (Å) | 1.0332 | 1.0332 |
| **Refinement** | | |
| Molecules/ASU | 4 | 2 |
| Resolution (Å) | 50.0-2.66 | 50.0-2.77 |
| | (2.73-2.66) | (2.84-2.77) |
| $R_{work}$ (%)[e] | 24.77 (42.74) | 25.14 (42.49) |
| $R_{free}$ (%)[f] | 27.96 (47.72) | 28.20 (42.03) |
| RMSD bond (Å) | 0.003 | 0.003 |
| RMSD angle (°) | 0.736 | 0.830 |
| Number of atoms | | |
| Protein[g] | 56,015 | 27,858 |
| Carbohydrate/metal ion | 1473 | 729 |
| Water | 431 | 203 |
| B-factors (Å$^2$) | | |
| Protein | 118.9 | 126.0 |
| Carbohydrate/metal ion | 121.1 | 131.4 |
| Water | 58.7 | 67.2 |
| Ramachandran (%)[h] | 91.93, 7.66, 0.41 | 92.84, 7.05, 0.11 |
| MolProbity percentile[h] | | |
| Clash/Geometry | 98/98 | 100/99 |
| PDB code | 6OM1 | 6OM2 |

[a]Integrin αVβ8 with αV residues 1–594 and M400C mutation and β8 residues 1–456 with V259C mutation
[b]Values within parentheses refer to the highest resolution shell
[c]$R_{merge} = \Sigma h \Sigma i \ |li(h) - <I(h)>|/\Sigma h\Sigma i \ li(h)$, where $li(h)$ and $<I(h)>$ are the ith and mean measurement of the intensity of reflection h
[d]Pearson's correlation coefficient between average intensities of random half-data sets for unique reflections[27]
[e]$R_{work} = \Sigma h||F_{obs} (h)| - |F_{calc} (h)||/\Sigma h|F_{obs} (h)|$, where $F_{obs} (h)$ and $F_{calc} (h)$ are the observed and calculated structure factors, respectively. No $I/\sigma(I)$ cutoff was applied
[f]$R_{free}$ is the R value obtained for a test set of reflections consisting of a randomly selected 1.4% (unliganded) and 2.6% (liganded) subset of the dataset excluded from refinement
[g]Among all independent unliganded and liganded structures, respectively, the average number of residues that could be built for PSI was 64 and 74% (β6) and 0 and 0% (β8), for hybrid was 100 and 100% (β6) and 71 and 61% (β8), for βI was 100 and 100% (β6) and 96 and 100% (β8), and for I-EGF-1 was 11 and 81% (β6) and 0 and 0% (β8)
[h]Calculated with MolProbity[30]

of an ADMIDAS metal ion in αVβ8. $Ca^{2+}$ binding is also competed by the hydrogen bond of Asn-120 to Gln-302 (Fig. 3a, b). Gln-302 in β8 replaces the Thr in the β5-α6 loop found in all other integrins (Fig. 3g).

**Ligand binding.** The $R^{215}GDLATI^{221}$ sequence motif in the TGF-β1 peptide binds to αVβ8 (Fig. 2e). The ligand Arg-215 sidechain hydrogen bonds to the αV Asp-218 sidechain. The ligand Asp-217 sidechain coordinates the MIDAS $Mg^{2+}$. Compared to unliganded β8, specificity-determining loop 1 (SDL1) at the beginning of the α1-helix with its MIDAS-coordinating residues moves toward the $Mg^{2+}$. This movement enables coordination of Ser-116, that is, the β-MIDAS S5 residue, to the MIDAS $Mg^{2+}$ and a hydrogen bond of the Asp-217 sidechain to the SDL1 backbone. The orientations of Arg-215 and Asp-217 sidechains are supported by hydrogen bonds of their backbones to αVβ8. The ligand LATI sequence has an α-helix-like conformation with its hydrophobic Leu-218 and Ile-221 in a pocket

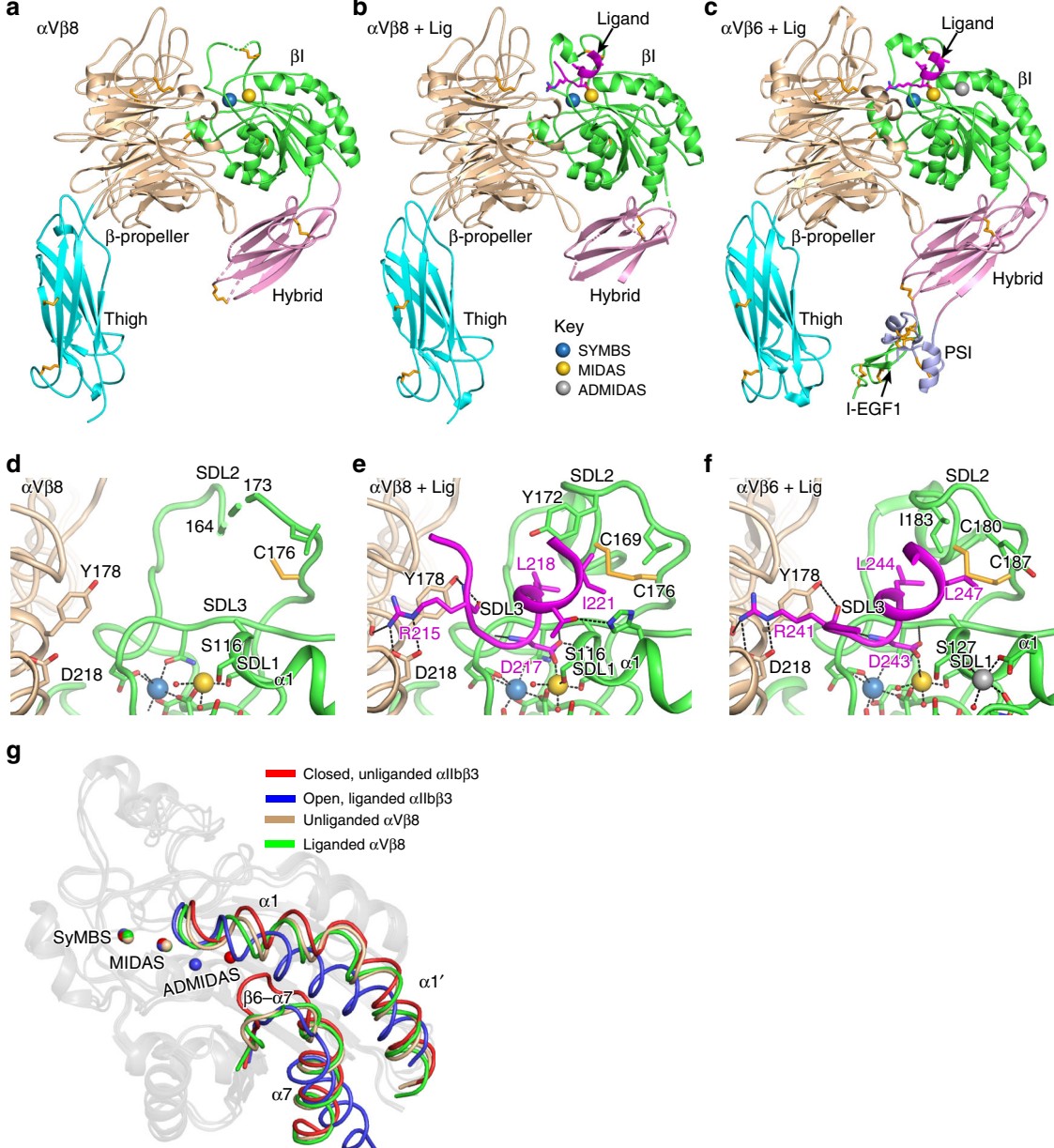

**Fig. 2** αVβ8 headpiece structure and ligand-binding site. **a–c** Overall headpiece structures and **d–f** ligand-binding sites of αVβ8 (**a**, **d**), αVβ8 with ligand (**b**, **e**), and αVβ6 with ligand (**c**, **f**)[17]. The color scheme in **d–f** is the same as in **a–c**. In αVβ8 the PSI and EGF-1 domains are missing in electron density as are portions of the hybrid domain; shorter missing breaks in the hybrid domain are dashed. In all panels, structure representation in PyMol shows ribbon cartoon, key sidechains with oxygens in red and nitrogens in blue, disulfides in yellow, metals in the βI domain as spheres, and metal coordination bonds and key hydrogen bonds as dashed lines. Waters are shown as small red spheres. **g** βI domain regions that move in allostery in typical integrins are compared to their counterparts in αVβ8 and shown in colored worm-like traces, while non-mobile regions are shown in gray ribbon cartoon. Metal ions are shown as spheres with the same color code as worm-like traces. Structures are closed, unliganded (PDB code 3T3P) and open, liganded (2VDR) αIIbβ3 and unliganded (chains A and B), and liganded (chains C and D) of αVβ8.

formed by the β8 SDL2 loop. In unliganded αVβ8, nine residues in the SDL2 loop, including one of the disulfide-bonded Cys residues, are disordered (Fig. 2d). Contact with hydrophobic ligand residues Leu-218 and Ile-221 contributes to SDL2 ordering, including of residue Tyr-172 (Fig. 2d, e).

The overall binding mode is similar to that of a TGF-β3 peptide bound to αVβ6 (Fig. 2f) and a lower resolution structure of dimeric pro-TGF-β1 bound to αVβ6[20]. However, there are important differences. In absence of ligand, the SDL2 loop of β6 is ordered, correlating with the presence of multiple backbone hydrogen bonds[17]. Furthermore, when ligand was soaked into αVβ8 crystals, Ser-116 in the βI α1-helix came into direct

coordination with the MIDAS $Mg^{2+}$ ion, while when αVβ6 crystals were soaked with ligand, the corresponding S5 residue, Ser-127 did not (Fig. 2e, f). These differences occur because during soaking, binding of ligand to αVβ8 induces far more movement of the S5 residue and the α1-helix that bears it than occurs in αVβ6 (Fig. 4). In αVβ8, the S5 residue Cα atom moves about 1.4 Å during soaking to a position that is within 1.1 Å of the position of S5 in fully open integrin αIIbβ3. In contrast, little movement of S5 occurs in integrin αVβ6 during soaking, since its position remains close to that observed in fully closed, unliganded integrins (Fig. 4). In contrast, when ligand is bound to αVβ6 first, and positions of its domains and especially its hybrid domain are

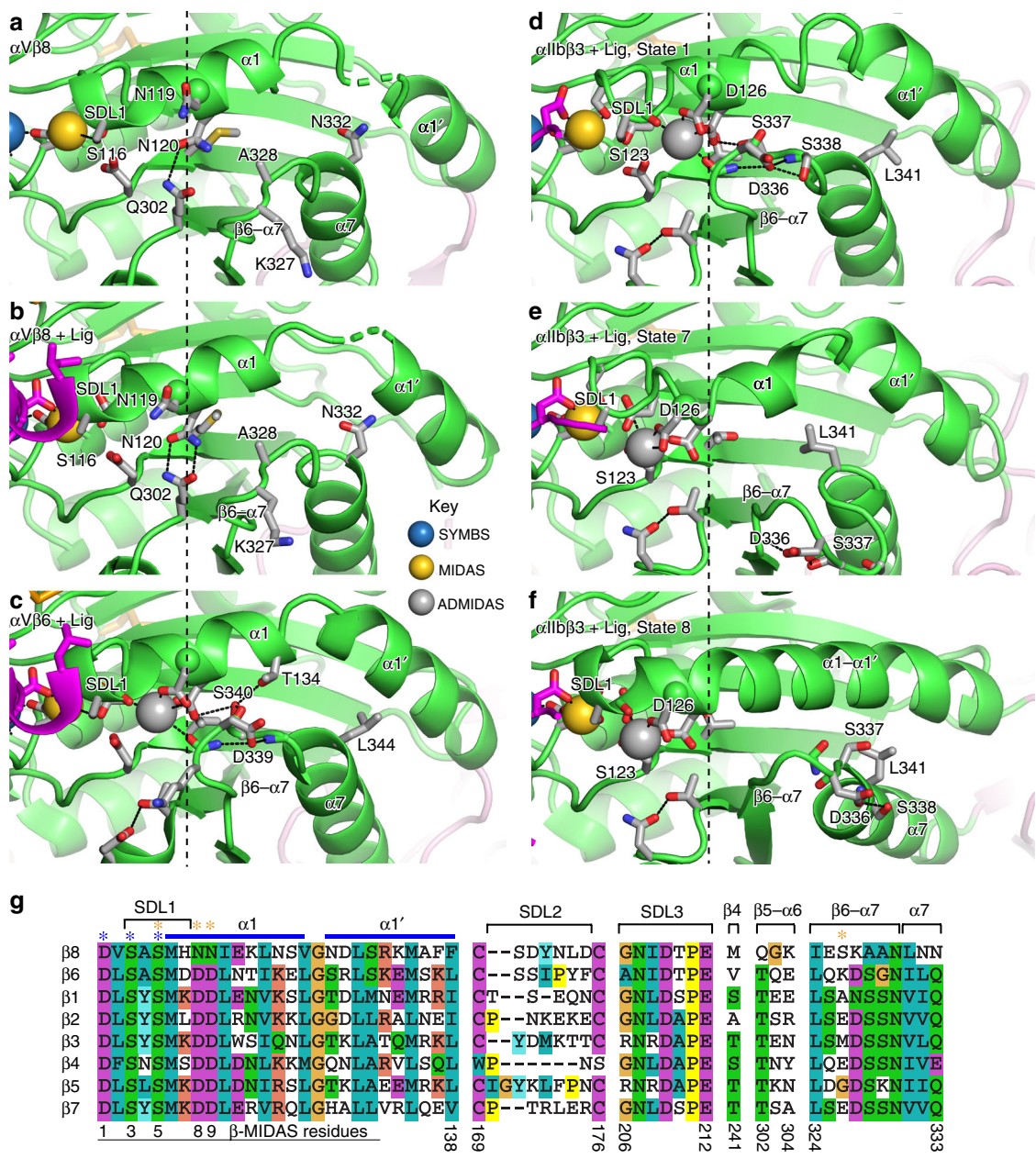

**Fig. 3** Residues important in βI domain allostery. **a–f** The mobile portion of the βI domain is shown in **a** unliganded αVβ8, chain D; **b** liganded αVβ8, chain B; **c** liganded αVβ6 (4UM9, chain B); **d** liganded αIIbβ3 in state 1 (3ZDY, chain D); **e** liganded αIIbβ3 in state 7 (3ZDZ, chain B); and **f** liganded αIIbβ3 in state 8 (2VDR, chain B). Structure representation is as in Fig. 2, except that sidechain carbons are in silver and mainchain carbons are in green. Vertical dashed lines mark the position of the β-MIDAS motif D8 Cα atom in αIIbβ3 state 1 (**d**). **g** Sequences of all human integrin βI domains in regions that are important in ligand binding, shape shifting, or appear to have unusual residues in β8. Residues with sidechain or mainchain coordination to the MIDAS or ADMIDAS in typical integrins are asterisked in blue and orange, respectively. β-MIDAS residue positions and β8-subunit residue numbers are shown at the bottom.

not restrained by crystal lattice contacts, its S5 residue moves 1.5 Å further to a fully open position and directly coordinates the MIDAS $Mg^{2+}$ ion (Fig. 3). This movement is expected to increase affinity because the ligand-binding pocket is tightened up, and it enables direct coordination of the S5 residue to $Mg^{2+}$, enhancing covalent-like coordination of $Mg^{2+}$ to ligand. Moreover, the energetic cost of inducing a change in structure of the integrin can only be paid thermodynamically if the intermediate structure has higher affinity for ligand than the closed structure.

**Ligand-induced shape shifting and unique αVβ8 features.** Soaking ligand into αVβ8 crystals induces movement toward the

ligand of the S5 and N8 residues and the α1-helix in which they locate to positions that are intermediate between closed and open (Figs. 3 and 4). Eight states along the conformational change pathway, including closed state 1, open state 8, and intermediate states 2–7, have been captured in αIIbβ3 crystal structures[21]. αIIbβ3 states 1, 7, and 8 are shown in Fig. 3d–f for comparison to αVβ8. Throughout the shape-shifting process, the α1-helix, with SDL1 at its tip (Fig. 3g), moves toward the ligand and tightens its binding pocket at the MIDAS. The ADMIDAS $Ca^{2+}$ ion moves with its coordinating D8 and D9 residues in the α1-helix. The position of the β-MIDAS D8 residue or N8 residue in β8 is marked with a Cα-sphere in Fig. 3a–f. For comparison among superimposed βI domains, the vertical dashed lines in

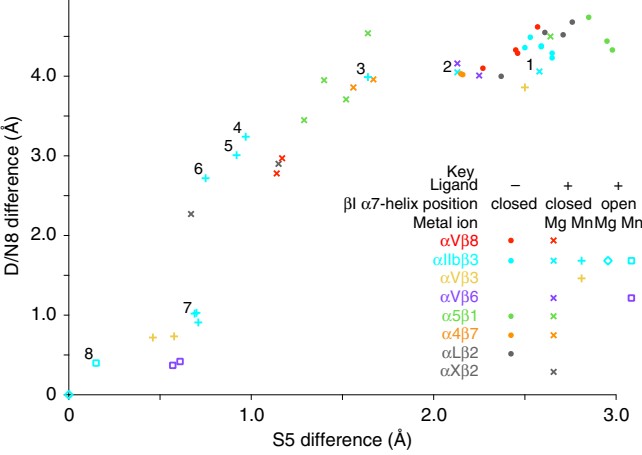

**Fig. 4** Measurements from crystal structures of the relative positions of β-MIDAS S5 and D/N8 residues. Distances from an open conformation structure (integrin αIIbβ3, PDB code 2VDR) of β-MIDAS S5 and D/N8 residues in liganded and unliganded integrin structures. Measurements show Cα atom—Cα atom distances. Numbers mark closed (1), open (8), and intermediate (2-7) ligand-bound states defined in a study in which ligand was soaked at different concentrations in $Mg^{2+}$ or $Mn^{2+}$ into crystals containing two independent copies of the closed αIIbβ3 headpiece[21]. Crystal structures with missing βI domain metals or lacking deposited structure factors were not plotted. Representative βI domains from each integrin β-subunit were superimposed with Deep Align[36]. Each independent molecule in asymmetric units was then superimposed on the cognate β-subunit and distances were measured with PyMol to one-hundredth Å. The PDB code and chain ID of the plotted structure models listed as PDB/Chain ID in order of increasing S5 difference are 2VDR/B, 3ZE2/D, 4MMY/B, 5FF0/F, 4MMX/B, 5FF0/B, 4NEH/B, 3ZE0/B, 3ZE1/B, 3ZDZ/B, 3ZE2/B, 3ZE1/D, 3ZE0/D, 6OM2/B, 4NEN/B, 6OM2/D, 4WK4/B, 3VI4/B, 4WK2/B, 3ZDZ/D, 3VI4/D, 3ZDY/B, 4UM9/B, 3V4P/B, 3V4P/D, 4UM9/D, 6OM1/B, 5ES4/B, 6OM1/H, 6OM1/F, 3FCS/B, 4MMZ/B, 3FCS/D, 6OM1/D, 3ZDY/D, 3T3P/B, 3ZDX/B, 5E6U/B, 4WK0/B, 3T3P/D, 3ZDX/D, 5E6R/B, 5E6S/B, 4WJK/B, 3VI3/D, 3VI3/B.

Fig. 3a–f mark the position of the state 1 αIIbβ3 D8 residue Cα atom sphere. Between closed state 1 and open state 8 in αIIbβ3 (Fig. 3d–f), the ADMIDAS metal ion moves 3.6 Å, the β-MIDAS D8 residue moves 3.5 Å, and the β-MIDAS S5 residue moves 2.3 Å (Fig. 4). In response to binding soaked-in TGF-β1 peptide, in αVβ8 the β-MIDAS D/N8 equivalent Asn-119 residue moves 1.8 Å and the β-MIDAS S5 Ser-116 residue moves 1.4 Å (Fig. 2g). Thus, although αVβ8 does not exhibit headpiece opening in electron microscopy (EM) and lacks an ADMIDAS, its β-MIDAS S5 residue moves substantially towards the ligand-binding pocket.

The β8 βI domain not only lacks an ADMIDAS but also displays differences from all previously structurally characterized integrin βI domains in the α1 and α1′ helices and the β6-α7 loop. β8 is exceptional for lacking electron density for residues between the βI domain α1 and α1′ helices in one of four different unliganded molecules and one of two different liganded molecules in crystal asymmetric units (Fig. 3a, b). Disorder at this position has not been seen in any of a large number of previous integrin crystal structures and appears to be related to an unusual conformation of the β8 βI domain β6-α7 loop that makes it unengaged with the α1 and α1′ helices, as discussed in the next paragraph. Furthermore, the α1′-helix in β8 differs in position by 1.5 Å from other integrins (Fig. 2g). None of these differences are at lattice contacts in the αVβ8 crystal structures.

β8 differs from typical integrins in lacking an (Asp/Asn)-Ser motif in the β6-α7 loop (Fig. 3g), which enforces a stereotypical

loop conformation in all previously crystallized integrins. The Asp sidechain of this motif (D339 in β6 and D336 in β3) hydrogen bonds to its own backbone nitrogen (at the 0 position) and that of the residue in the +2 position to stabilize the turn between the β6-strand and α7-helix (Fig. 3c, d). The Ser residue often present in the +2 position (S338 in β3) further stabilizes the turn by also hydrogen bonding to the Asp (Fig. 3d). Importantly, the hydrogen bonds to the β6-α7 peptide backbone stabilize the orientation of the carbonyl oxygen of the residue in the −1 position, which coordinates to the ADMIDAS metal ion (Fig. 3c, d). The Ser residue in the +1 position of the (Asp/Asn)-Ser motif of typical integrins hydrogen bonds to an ADMIDAS-coordinating Asp, and, in some integrins, also to other residues in the α1-helix such as Thr-134 in β6 (Fig. 3c). Finally, in place of hydrophilic β8 Asn-332 (Fig. 3a, b), all other integrins contain a hydrophobic Leu, Val, or Ile residue (Fig. 3g), which stabilizes interaction with the α1′-helix (Leu-344 β6 and Leu-341 in β3) (Fig. 3c–f). Thus, the β6-α7 loop in non-β8 integrins has a specific hydrogen bond-stabilized conformation and sequence of amino acid sidechains that are integral to promoting multiple interactions of the β6-α7 loop with the ADMIDAS, α1-helix, and α1′-helix.

In place of the Asp/Asn residue at the tip of the β6-α7 loop in typical integrins (Fig. 4g), Lys-327 in β8 locates 6 Å more distal from the α1-helix (Fig. 3a–d). The lack of restraining interactions with the β6-α7 loop is expected to enable greater movement of the β1-α1 loop and α1-helix toward bound ligand in β8 than in typical integrin β-subunits. In typical integrins, the β6-α7 loop, and particularly the Ser of the (Asp/Asn)-Ser motif, hinders α1-helix movement toward ligand and the pivoting movement of the α1′-helix when it fuses with the α1-helix (Fig. 3, panel f compared to a–e). In states visualized in αIIbβ3 crystals, this Ser, Ser-337, moves only 0 to 0.5 Å from state 1 to state 6, but 11 Å in state 7, and 7 Å in open state 8 (Fig. 3d–f).

It appears that the position of the βI domain β6-α7 loop in β8 would accommodate tilting of the α1′-helix and its fusion with the α1-helix, without requiring pistoning of the α7-helix and swing-out of the hybrid domain as seen in typical integrins. The unique features of the β8-subunit may enable intermediate or even complete movement of SDL1 toward the open conformation to be dissociated from pistoning of the α7-helix and swing-out of the hybrid domain (Fig. 1f). These features may be responsible for the finding that in contrast to other integrins, when αVβ8 binds ligand, headpiece opening as assessed by hybrid domain swing-out is not visualized in EM[14,15].

**HDX mass spectrometry.** HDX mass spectrometry (MS) on the αVβ6 and αVβ8 headpieces showed similar backbone dynamics of their αV-subunits including slow exchange in the β-propeller domain and interesting differences in dynamics of their β6- and β8-subunits. Multiple peptides covering the βI domain α1 and α1′ helices generally showed more rapid exchange of backbone amide hydrogens in β8 than β6 (Fig. 5 and Supplementary Figs. 1 and 2), although these differences cannot be accurately quantified given that the sequences of β6 and β8 are not identical and totally deuterated proteins for control studies could not be prepared. Nevertheless, the results show that the α1–α1′ region is generally more mobile in β8 than in β6, in agreement with disorder of α1–α1′ helix residues in the β8 structure.

The effect of ligand binding on αVβ6 and αVβ8 was also examined by HDX MS. In the αV β-propeller domain, Tyr-178 and Asp-218 bind Arg-215 of TGF-β1 (Fig. 2e). In the C-terminal half of the region between these residues, ligand binding slowed the exchange of peptides in both integrins (Fig. 6a, c). Three regions were affected in the βI domains of both integrins

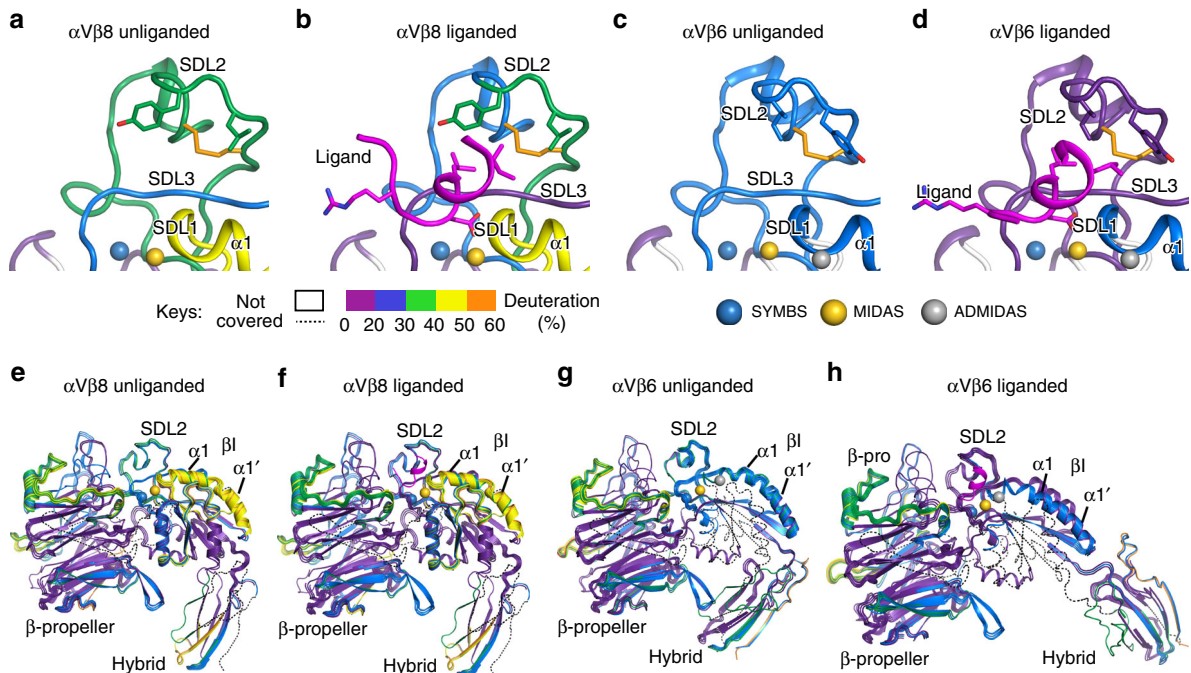

**Fig. 5** Hydrogen-deuterium exchange dynamics. HDX of αVβ8 and αVβ6 headpiece fragments in presence or absence of TGF-β1 ligand peptide at 1 min. (**a–d**). Close-ups of the βI domain around the ligand-binding site. (**e–h**). The β-propeller, hybrid, and βI domains. Cartoon diagrams are colored according to the key for deuterium exchange at 1 min. for a single set of non-overlapping peptides (a-d) or for all peptides (e-h), except TGF-β1 peptide is shown in magenta. As HDX data covers regions disordered in crystal structures, the structure of SDL2 from liganded αVβ8 and the hybrid domain from αVβ6 are used to model disordered regions of αVβ8; furthermore, the hybrid domain of αVβ6 swings out in presence of ligand and its position is modeled on αIIbβ3.

(Fig. 6b, d–h). We chose a 1 Da cutoff to mark HDX differences that are clearly above noise and are likely meaningful structurally. Other changes in the range of 0.5–1.0 Da are above triplicate variation and may also have limited importance. In αVβ8, the relatively rapid exchange of the α1-helix was augmented at 10 s by ligand binding (Fig. 6b). In αVβ6 by contrast, exchange in the α1-helix was decreased at early time points by ligand binding (Fig. 6d). In SDL2, ligand binding greatly decreased exchange in both integrins, in agreement with the ordering of SDL2 in αVβ8 (Fig. 6b, d). In the absence of ligand, generally faster exchange of SDL2 in β8 than β6 was in agreement with disorder in β8 and not β6 crystal structures (Figs. 5a, d and 6e, g). Finally, ligand binding slowed exchange in peptides encompassing SDL3 between the α2 and α3 helices in both integrins (Figs. 5a–d and 6b, d–g). SDL3 underlies the ligand-binding site in both αVβ6 and αVβ8 and forms multiple hydrogen bonds to the ligand RGD moiety (Fig. 2e, f). Detailed comparisons between αVβ6 and αVβ8 are not possible in SDL2 and SDL3 because in addition to the effects of sequence differences on rates of exchange, the lengths of the peptides and the positions of their midpoints plotted in Fig. 6b, d varies, as shown by plotting the peptides (Fig. 6e–h).

**Unique features in αVβ8 regulate ligand-binding affinity.** To test the significance of structural differences associated with specific sequence differences in β8 compared to other integrins, we investigated their effect on ligand-binding affinity and the ability of $Mn^{2+}$ to augment this affinity. Residues in shape-shifting interfaces within the βI domain engaged in distinctive interactions in β8 and β6 were exchanged, including those in the α1 and α1′ helices and the β5-α6 and β6-α7 loops (Figs. 3g and 7a). Over the entire ectodomain, β8 and β6 are 40% identical, and identity is highest in the βI domain, at 48%. Affinities were measured by fluorescence polarization by binding to fluorescently labeled pro-TGF-β1 peptide in solution (Fig. 7b and

Supplementary Fig. 3). Introducing all 19 β6 residues into β8 in αVβ8-mut5 lowered affinity in $Mg^{2+}$ and $Mn^{2+}$ by 5- and 3-fold, respectively. Surprisingly, exchange of three α1-helix residues including NN to DD at the ADMIDAS had no significant effect (αVβ8-mut6, Fig. 7a). Furthermore, exchange of both Asn residues (αVβ8-mut8) or one Asn residue plus the Thr residue found in all integrins except β8 in the β5-α6 loop (αVβ8-mut1) raised affinity in $Mg^{2+}$ by 2-fold (Fig. 7a). Because αVβ8-mut3 showed a greater increase in affinity in $Mn^{2+}$ (4-fold) than wild-type (WT) (2-fold) and αVβ8-mut6 exchanged ADMIDAS residues thought to be important in headpiece opening, these mutants were tested for headpiece opening in presence of ligand and $Mn^{2+}$. Negative stain EM showed that like αVβ8-WT, and unlike αVβ6-WT, the headpiece of αVβ8-mut3 and αVβ8-mut6 remained closed when bound to pro-TGF-β1 in $Mn^{2+}$ (Fig. 7c and Supplementary Fig. 4). The role of the ADMIDAS Asn residues in αVβ8 was further tested by mutation to alanine. αVβ8-mut7 showed a 7- and 3-fold decrease in affinity in $Mg^{2+}$ and $Mn^{2+}$, respectively (Fig. 7a).

We further measured the importance of the residues in shape-shifting regions more C-terminal than the ADMIDAS. Although residues Phe-137 and Phe-138 at the C-terminus of the α1′-helix are bulkier and more hydrophobic than those in other integrins (Fig. 3g), their exchange in αVβ8-mut4 only modestly increased affinity relative to WT, consistent with the difference in affinity between αVβ8-mut3 and αVβ8-mut5, which differ by the same two residues. Mutant αVβ8-mut2 exchanged residues in the β4-strand, β5-α6 and β6-α7 loops, and α7-helix (14 residues including 7 in the β6-α7 loop and 3 in the α7-helix). αVβ8-mut9 and mut10 exchanged only two and three residues in the β6-α7 loop, respectively. All three mutants showed similar decreases in affinity of 2- to 3-fold in both $Mg^{2+}$ and $Mn^{2+}$ (Fig. 7a). Overall, the results show that residues that are unique to β8 compared to all other integrins and are in regions that change shape between the unliganded and liganded states of αVβ8 are

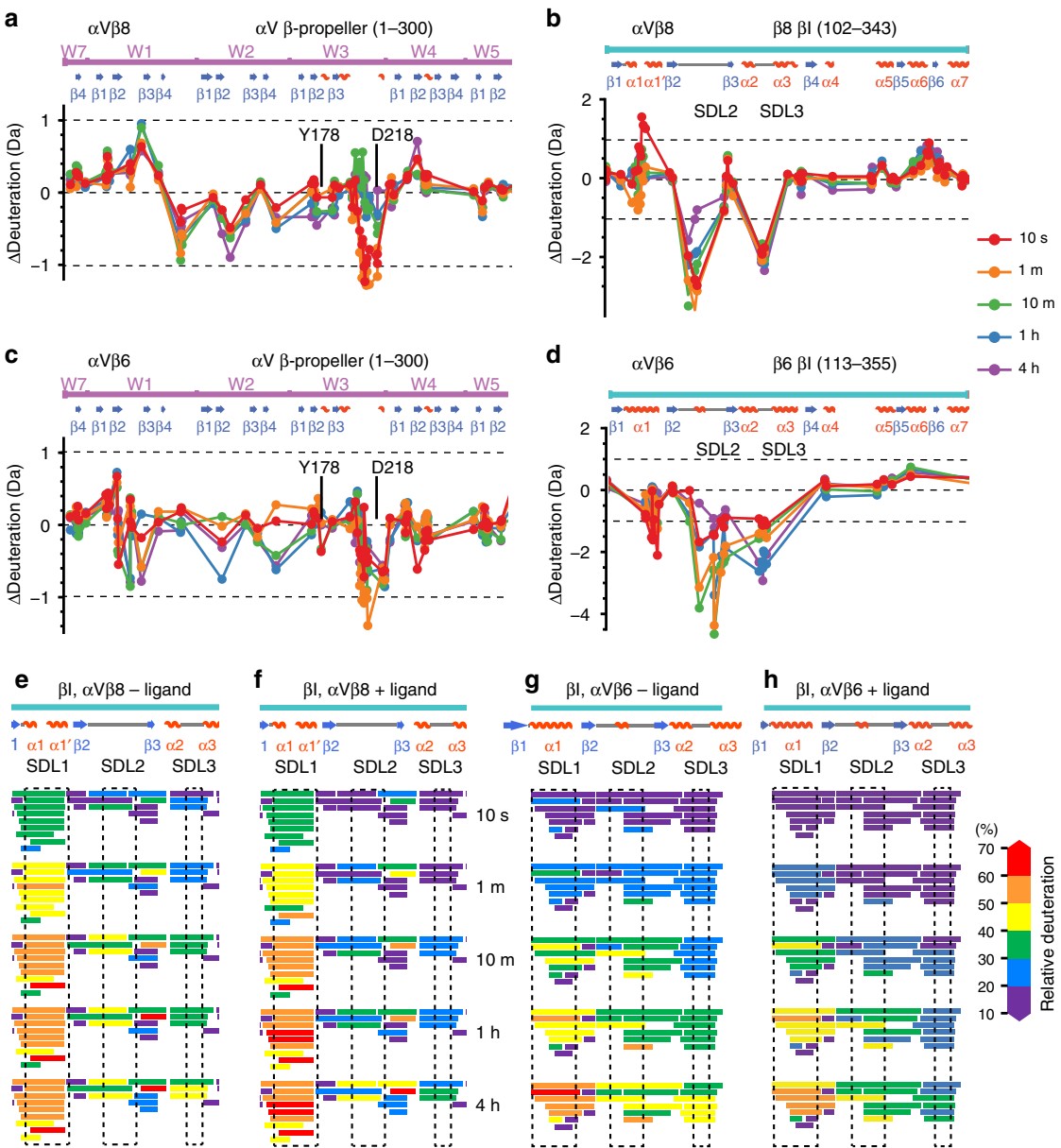

**Fig. 6** Effect of ligand binding on deuterium exchange. **a–d** Differences in HDX with and without saturating concentrations of TGF-β1 ligand peptide G[213]RRGDLATIHG[223] for αVβ8 and αVβ6 are shown for each peptide plotted at the midpoint of its sequence position for a portion of the β-propeller domain and the entire βI domain. The equation for subtraction was ($D_{liganded} - D_{unliganded}$. Differences > 1 Da (dashed lines) are considered meaningful. All HDX data are presented in Supplementary Figs. 1 and 2. **e–h** Details of the ligand-binding region of the βI domain. Exchange in each peptide is colored according to the key.

important for the ability of αVβ8 to bind ligand with high affinity in the absence of headpiece opening. Surprisingly, we also found that residues in the β6-α7 loop made an important contribution to the specialization of αVβ8 to bind ligand with high affinity, in correlation with the unique, unengaged structure of the β6-α7 loop in β8.

Most interestingly, while many substitutions to β6 residues lowered affinity of αVβ8, many substitutions to β8 residues raised affinity of αVβ6. Affinity of the αVβ6-NN mutant was increased 6-fold, suggesting that lack of an ADMIDAS metal ion may enable greater shifting toward the open conformation of the βI domain in the absence of hybrid domain swing-out (Fig. 7a). Even more dramatically, replacing the DS motif in the β6-α7 loop of αVβ6 with β8 sequence in the αVβ6-DS mutation increased

affinity 13-fold, demonstrating the previously unsuspected importance of this loop for maintaining the low-affinity state.

## Discussion

Integrin αVβ8 has multiple differences from typical integrins that may relate to its distinctive, non-talin-dependent activation mechanism. In typical integrins, the D8 and D9 β-MIDAS residues provide the only sidechains that coordinate the ADMIDAS $Ca^{2+}$ ion. Their substitution with Asn in β8 results in a lack of a bound metal ion, and different sidechain orientations. The β8 β-MIDAS N9 Asn residue forms a sidechain–sidechain hydrogen bond to Gln-302 in the β5-α6 loop, at which position all other integrins have a Thr residue (Fig. 3g). We examined

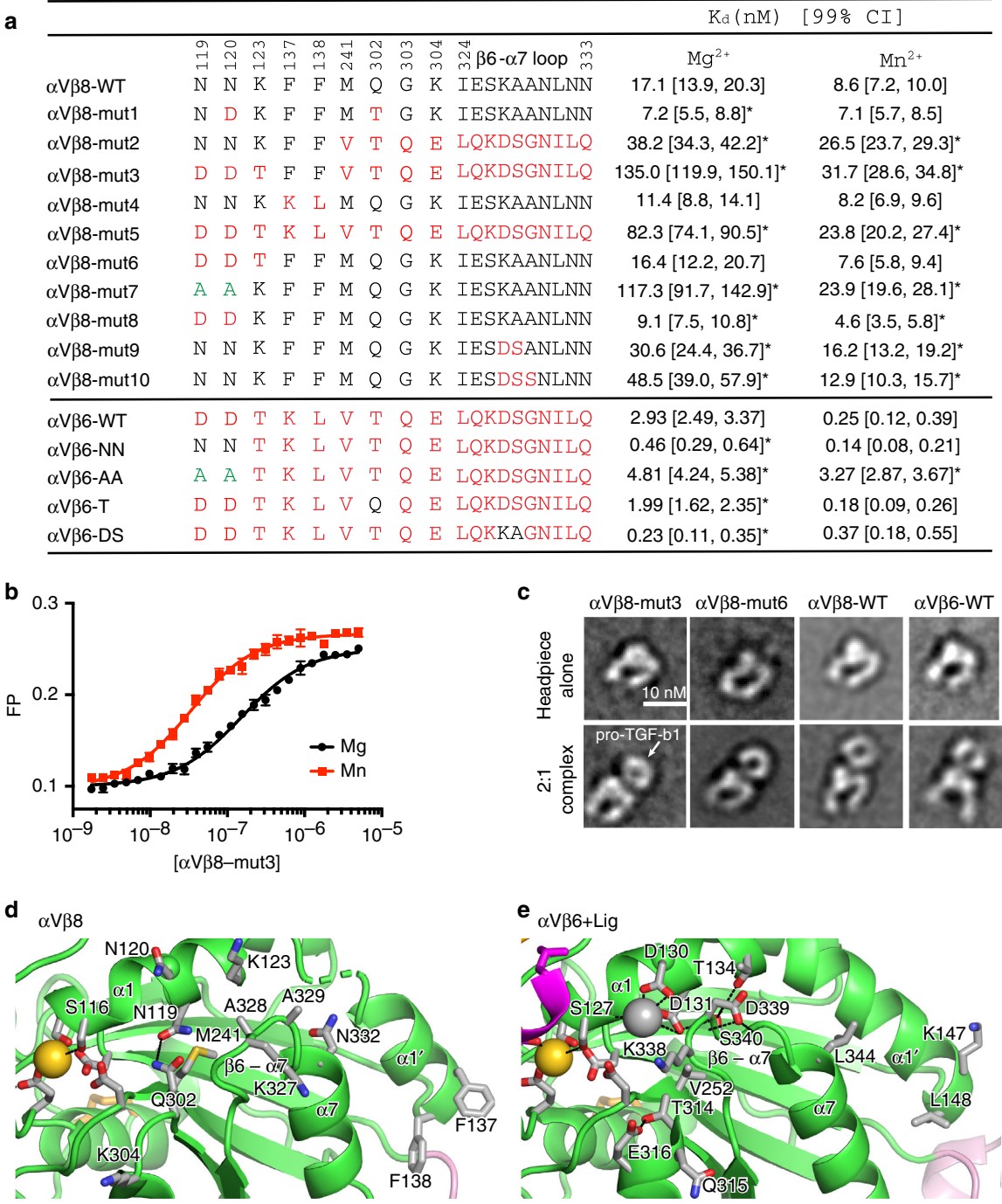

**Fig. 7** Regulation of ligand-binding affinity by atypical and typical residues in αVβ8 and αVβ6. **a** Mutations corresponding to sequence exchanges between αVβ8 (black) and αVβ6 (red) and effect on affinity for FITC-labeled TGF-β3 ligand peptide (GRGDLGRLKK) in presence of $Mg^{2+}$ and $Mn^{2+}$. $K_D$ and [99% confidence interval] values were determined by fluorescence polarization and fits using the NonLinearModelFit function of Mathematica (Wolfram, Champaign, IL) of all data from measurements in triplicate of two different experiments done in different months. As in statistical models for determining $p$ values, the fit assumes that errors are independent and normally distributed. The reported fit minimizes the sum of the squared errors. If two values have 99% confidence intervals that do not overlap, then the two values by definition have <1% chance of being different by chance alone, that is, the $p$ value is <0.01. Therefore, mutant $K_D$ values with confidence intervals that do not overlap with those of WT are significantly different ($p < 0.01$) and are asterisked. **b** Representative fluorescence polarization (FP) of αVβ8-mut3 from one triplicate experiment (average ± s.d) with fit (line). Source data for **a**, **b** are provided as a Source Data file. **c** Representative negative stain class averages of mutant or WT integrin headpieces alone (upper panels) or complexed with pro-TGF-β1. Representative 2:1 pro-TGF-β1:integrin class averages are shown. The WT integrin class averages shown for comparison are previously published[14]. **d**, **e** Residues in regions of βI domain allostery with unusual properties in αVβ8 (**d**) are compared to counterparts in αVβ6 (**e**).

conservation of N8 (Asn-119), N9 (Asn-120), and Gln-302 in integrin β8 in evolution. All three residues are invariant in mammals and chicken. Among fish (zebrafish, Japanese rice fish, spotted gar, and elephant shark), only Asn-120 (N9) is invariant. Asn-119 (N8) is found as Asp (twice), Glu, and Ala. Gln-302 is found as Asp (twice), Glu, and Gln. The sidechains of all of these residues at position 302 would be capable of hydrogen bonding to the invariant Asn-120 residue at position N9.

In typical integrins, the key process in raising integrin affinity for ligand during opening is movement of SDL1 in the α1-helix toward the ligand and the MIDAS $Mg^{2+}$ ion. Movement brings the Ser S5 β-MIDAS residue into direct coordination with the MIDAS metal ion, increases hydrogen bonding of the ligand Asp sidechain to the β1-α1 loop and α1-helix backbone, and tightens the ligand-binding pocket. Lessened exposure to solvent and the network of hydrogen bonds formed around the partially covalent ligand-$Mg^{2+}$ coordination bond increases its strength as explained in the Ligand-binding section of Results. We found that soaking ligand into crystals of αVβ8 induced substantial movement of SDL1 and the α1-helix toward the ligand, coordination of the S5 serine sidechain with the MIDAS $Mg^{2+}$, and hydrogen bonding of the SDL1 backbone to the ligand Asp sidechain. This liganded state of αVβ8 is intermediate between closed and open. In terms of the extent of movement toward the open state, wide variation is seen among integrins in crystals that are soaked with ligand. As shown with the talin-binding integrin αIIbβ3, the addition of $Mn^{2+}$ greatly increases the extent of movement toward the open state induced by soaking with ligand[21] (Fig. 4). Among crystallized integrins soaked with ligand in $Mg^{2+}$, αVβ8 shifts more than any other integrin (Fig. 4), consistent with its specialized features that enable increased affinity without hybrid domain swing-out, as demonstrated here by mutagenesis and EM.

In α4β1 and α5β1 integrins, complete transition from closed to open increases affinity for ligand by 700- and 4000-fold, respectively[9]. In αIIbβ3 crystals formed in the absence of ligand with the headpiece in the closed conformation, with two independent molecules in the crystal asymmetric unit, soaking in 10 mM RGD ligand in $Mg^{2+}/Ca^{2+}$ resulted in no conformational change in one molecule that remained in state 1 (closed) and a slight shift in the other to state 2[21]. In 1 mM RGD ligand in $Mn^{2+}/Ca^{2+}$, greater shift to state 3 occurred, and as ligand was increased from 3 to 5 to 10 mM in $Mn^{2+}/Ca^{2+}$, shifting gradually increased until reaching state 6 and finally, state 8 (open) (Fig. 4). Movement of the Ser S5 βMIDAS residue occurred throughout this process. The finding that ligand drives conformational change along this structural continuum, and that the degree of conformational change is dependent on ligand concentration (at least between states 3 and 8, which were all in $Mn^{2+}$), shows that affinity for ligand increases along the same continuum. Additionally, only the increase in affinity for ligand can pay the energetic cost required for the structural shifts within the integrin and the crystal lattice.

In typical integrins, coordination of the ADMIDAS $Ca^{2+}$ ion to the backbone carbonyl oxygen of the S5 β-MIDAS residue strongly restrains movement of this key residue and thereby hinders tightening of the ligand-binding pocket including formation of the direct coordination between the S5 Ser sidechain and the MIDAS Mg ion and the hydrogen bonds between the ligand Asp sidechain and the SDL1 backbone. The restraint is provided by the D8 and D9 β-MIDAS residues and the backbone carbonyl oxygen in the β6-α7 loop to which the ADMIDAS $Ca^{2+}$ ion also coordinates. This hypothesis was verified in plots of S5 and D/N8 β-MIDAS residue Cα atom positions relative to those in an open structure among 45 independent examples of integrin structures (Fig. 4). S5 and D8 residues do not shift proportionally

to one another, but rather show a sigmoid relationship. S5 moves relatively more in the early stages of opening, and then D8 catches up after the β6-α7 loop moves and its coordination to the ADMIDAS $Ca^{2+}$ ion is lost in state 7[21]. The lack of these restraints in the atypical β8-subunit is predicted to make the S5 β-MIDAS residue freer to move. It is not possible to verify this proposal from the measurements shown in Fig. 4, because crystal lattices and the conditions of soaking including ligand concentration and use of $Mn^{2+}$ vs. $Mg^{2+}$, as well as integrin structural features, may all influence the extent of shape shifting. Nonetheless, αVβ8 shifts more during soaking with ligand in $Mg^{2+}$ than any other integrin yet tested, that is, αVβ6, αIIbβ3, α5β1, and α4β7 (Fig. 4). αXβ2, also shown in Fig. 3, is not a direct comparison, because it was not soaked with ligand; it crystallized bound to its internal ligand, which in αI integrins binds to the same site to which αI-less integrins such as αVβ8 bind their "external" ligands.

In addition to the lack of an ADMIDAS, distinctive features of αVβ8 in the α1 and α1′ helices and β6-α7 loop are also expected to favor movement of the S5 β-MIDAS residue and the α1-helix toward the high-affinity state. Some examples of αVβ8 in crystals had missing electron density in the region between the βI domain α1 and α1′ helices, showing high flexibility. Furthermore, HDX showed that peptides encompassing the α1 and α1′ helices in the βI domain were much more flexible in αVβ8 than in αVβ6. One cause of this unusual flexibility is likely to be the unique position of the β6-α7 loop in the β8 βI domain. All integrin β-subunits but β8 have a (D/N)SXN motif in the β6-α7 loop (Fig. 3g). The first residue in this motif is at the tip of the β6-α7 loop, and an oxygen in its Asp/Asn sidechain hydrogen bonds to two backbone NH groups in the loop. These hydrogen bonds stabilize a highly specific conformation of the loop that keeps it close to the α1-helix until the final stage of opening when the β6-α7 loop reshapes and moves away (state 7 in Fig. 4). This movement permits α1 and α1′ helix merger and C-terminal pistoning of the α7-helix with hybrid domain swing-out in open state 8.

Two of the mutants with the greatest introduction of β6 residues into the β8-subunit were tested for hybrid domain swing-out. In contrast to results with αVβ6, neither mutant showed the open headpiece when bound to pro-TGF-β1. Our structure of αVβ8 shows that when ligand is soaked in, the βI domain shifts to a state intermediate between closed and open, as observed for typical integrins. Thus far, typical integrins show complete headpiece opening when co-crystallized with ligand, and when observed by EM when bound to ligand. Our results show that substitution with a set of up to 19 putatively atypical β8 residues with typical integrin residues in the βI domain was not sufficient to enable headpiece opening. We were unable to test whether lack of hybrid domain swing-out in β8 was intrinsic to its βI domain. Poor expression of β-subunit chimeras with βI domains swapped between β8 and β6 suggested structural incompatibilities. The βI-hybrid interface in β8 is typical in size and does not have an unusual number of hydrogen bonds (Supplementary Fig. 5).

Substitution of most residues from β6 for atypical residues in β8 lowered αVβ8 affinity for TGF-β1 peptide and substitution of atypical residues from β8 for typical residues raised αVβ6 affinity for TGF-β1 peptide. These findings suggest that the atypical residues in β8 enable greater movement of SDLI toward the ligand when ligand is bound than in typical integrins. The movements in β8 might correspond to greater movement to an intermediate conformation than is possible in typical integrins in the absence of hybrid domain swing-out, and in the absence of crystal lattice restraints, might extend to complete opening of the βI domain in the absence of hybrid domain swing-out.

$Mn^{2+}$ increases affinity of αVβ6 in part by stabilizing headpiece opening[12]. $Mn^{2+}$ boosted affinity of native αVβ6 by 12-fold

and of $\alpha V\beta 8$ by 2-fold, correlating with the lack of hybrid domain swing-out in $\alpha V\beta 8$. Replacing D8 and D9 in $\alpha V\beta 6$ with N8 and N9 increased affinity in $Mg^{2+}$ by 6-fold and decreased responsiveness to $Mn^{2+}$ to 3-fold compared to 12-fold in WT. This result is compatible with restraint of $\alpha 1$-helix/SDL1 movement by ADMIDAS $Ca^{2+}$ coordination and activation by $Mn^{2+}$ by replacement of $Ca^{2+}$ at the ADMIDAS[22]. Results with $\alpha V\beta 8$ were more complex. Substitution of one or both Asn with Asp increased affinity by 2-fold. However, when a nearby K123T mutation was added (DDT), there was no affinity increase compared to WT ($\alpha V\beta 8$-mut6). Furthermore, DDT + VTQE + LQKDSGNILQ ($\alpha V\beta 8$-mut3) was 3.5-fold lower in affinity than VTQE + LQKDSGNILQ ($\alpha V\beta 8$-mut2, Fig. 7a), rather than equal in affinity as expected from the equal affinities of DDT ($\alpha V\beta 8$-mut6) and WT. The effects of N in the D8 and D9 positions are thus dependent on the nature of residues in other positions in the $\beta I$ domain shape-shifting interface. Measurement of the amount of vitronectin binding to cells, rather than affinity, with $\alpha V\beta 3$ and $\alpha V\beta 8$ showed similar binding with WT and N8/N9 $\alpha V\beta 3$ and decreased binding of D8/D9 $\alpha V\beta 8$ compared to WT[23]. The reasons for these differences with the results here on pro-TGF-$\beta 1$ peptide affinity for $\alpha V\beta 8$ are unclear and are unlikely to be related to differences between use of headpiece versus intact integrins because the presence of the legs and TM domains are unlikely to have an effect on conformational equilibria in the absence of the conformations in $\alpha V\beta 8$ where they make a difference, that is, the bent-closed and extended-open conformations[9].

The importance of having either Asn or Asp at the $\beta$-MIDAS 8 and 9 positions was revealed by decease in affinity of both $\alpha V\beta 6$ and $\alpha V\beta 8$ after mutation to alanine. In the open conformation of typical integrins, the ADMIDAS $Ca^{2+}$ ion coordinates the Asp in the $\beta 4$-$\alpha 5$ loop and stabilizes its position in the outer coordination shell of the MIDAS $Mg^{2+}$ ion[20,24]. Perhaps, in the absence of an ADMIDAS $Ca^{2+}$ ion, a hydrogen bond with the N8 Asn in $\beta 8$ or the N8 Asn in N8/N9 mutant $\alpha V\beta 6$ provides a similar stabilizing role.

The conformation of the $\beta 6$-$\alpha 7$ loop in $\alpha V\beta 8$ is unique compared to the structure of this loop in the integrin $\beta 1$, $\beta 2$, $\beta 3$, $\beta 6$, and $\beta 7$ subunits. These represent five of the six talin/kindlin-binding integrin $\beta$-subunits and 21 of the 22 talin/kindlin-binding integrin $\alpha\beta$ heterodimers. The conformation of the $\beta 6$-$\alpha 7$ loop is essentially identical in the closed conformations of the latter integrins. This highly conserved $\beta 6$-$\alpha 7$ loop conformation is explained here by our observation of a (D/N)S motif with the Asp or Asn sidechain hydrogen bonding to two adjacent backbone amides to stabilize the tip of the $\beta 6$-$\alpha 7$ loop. Importantly, the $\beta 6$-$\alpha 7$ loop in the closed conformation packs against the $\alpha 1$-helix and the backbone carbonyl oxygen of the residue immediately preceding the (D/N)S motif coordinates the ADMIDAS $Ca^{2+}$ ion. Both interactions stabilize the SDL1/$\alpha 1$-helix position in the low-affinity, closed conformation.

In typical integrins, complete movement of the SDL1/$\alpha 1$-helix to the open state is only allowed when the $\beta 6$-$\alpha 7$ loop moves away from the $\alpha 1$-helix and toward the hybrid domain, which makes way for tilting of the $\alpha 1'$-helix and its fusion to the $\alpha 1$-helix and is accompanied by pistoning of the $\alpha 7$-helix toward the hybrid domain and swing-out of the hybrid domain, giving the open headpiece conformation (Fig. 1a–c). In $\beta 8$, the $\beta 6$-$\alpha 7$ loop is distal from the $\alpha 1$-helix, like the $\beta 6$-$\alpha 7$ loop in intermediate state 7 of $\beta 3$ and open state 8 of $\beta 3$ and $\beta 6$, and in a position where complete SDL1/$\alpha 1$-helix movement and $\alpha 1'$-helix tilting and merger appear possible without $\alpha 7$-helix pistoning and headpiece opening (Fig. 1d–f).

Mutations verified an important role for the $\beta 6$-$\alpha 7$ loop in regulating affinity of both $\alpha V\beta 6$ and $\alpha V\beta 8$. Replacing the DS motif of $\beta 6$ with KA from $\beta 8$ resulted in a 13-fold increase in affinity of $\alpha V\beta 6$. Conversely, replacing KA with DS or KAA with DSS

decreased $\alpha V\beta 8$ affinity by 1.8- and 2.8-fold, respectively. These results demonstrate an important and previously unexpected role of the $\beta 6$-$\alpha 7$ loop in regulating affinity of both typical integrins and atypical integrin $\alpha V\beta 8$. The results suggest that in typical integrins, the DS motif maintains a conformation of the $\beta 6$-$\alpha 7$ loop that restrains SDL1/$\alpha 1$-helix movement, whereas the lack of this motif and the unengaged conformation of the $\beta 6$-$\alpha 7$ loop are permissive of SDL1/$\alpha 1$-helix movement toward the open conformation.

Our structures of liganded and unliganded $\alpha V\beta 8$ suggest that tightening of the ligand-binding pocket in $\alpha V\beta 8$ with MIDAS-proximal movement of SDL1 to an open-like conformation is plausible in the absence of hybrid domain swing-out. In the open $\beta I$ domain conformation of typical integrins, the $\alpha 1'$-helix pivots and invades the space occupied by the $\beta 6$-$\alpha 7$ loop in its closed conformation. Because the $\beta 6$-$\alpha 7$ loop in $\beta 8$ is atypically distal from the $\alpha 1$ and $\alpha 1'$ helices, space is available for $\alpha 1'$-helix pivoting and fusion with the $\alpha 1$-helix. Together with the lack of ADMIDAS metal coordination bonds, the altered $\beta 6$-$\alpha 7$ loop creates more freedom to accommodate $\alpha 1$-helix movement to its conformation in the open $\beta I$ domain. Our studies explain why ligand binding does not induce headpiece opening of $\alpha V\beta 8$. However, as none of the unusual features of the $\beta 8$ $\beta I$ domain would be near the integrin lower legs in the bent integrin conformation, they do not explain the low abundance of the bent conformation for $\alpha V\beta 8$.

The structure and mutagenesis results reported here suggest that the $\beta 8$ $\beta I$ domain may open in the absence of hybrid domain swing-out. The atypical structural features of the $\beta 8$ $\beta I$ domain, including the disengagement of the $\beta 6$-$\alpha 7$ loop from the $\alpha 1$-helix, suggest that complete movement of SDL1 with the $\alpha 1$ and $\alpha 1'$ helices toward the ligand and $\alpha 1$ and $\alpha 1'$ helix merger, resulting in high affinity for ligand, may occur without requiring the $\alpha 7$-helix pistoning and hybrid domain swing-out that is seen in typical integrins (Fig. 1c, f). Thus, $\alpha V\beta 8$ appears to have two states, one of which is found in typical integrins, extended-closed, and another which appears unique, extended with a $\beta I$ domain in which the $\alpha 1$-helix is in a partially or fully open position and the $\alpha 7$-helix and hybrid domain are in a closed position.

The unique conformational ensemble of $\alpha V\beta 8$ correlates with its unique linkage among integrins not to talin but to Band 4.1. We have revealed in $\alpha V\beta 8$ the striking absence of an ADMIDAS, a more mobile SDL1/$\alpha 1$ and $\alpha 1'$ helix, and a $\beta 6$-$\alpha 7$ loop that is disengaged from the $\alpha 1$-helix. Moreover, our demonstration that a (D/N)S motif in the $\beta 6$-$\alpha 7$ loop stabilizes typical integrins in their low-affinity state revealed insights into the structural principles that regulate activation not only of atypical integrin $\alpha V\beta 8$ but also of typical integrins. Distinct structural features in $\beta I$ domains may tune them to be activated by adaptors that couple to different cellular cytoskeletal systems.

## Methods

**Proteins**. An $\alpha V\beta 8$ headpiece construct with an $\alpha V$-$\beta 8$ disulfide was prepared and expressed in HEK293S GntI$^{-/-}$ cells exactly as previously described[14]. Cells were obtained from the authors[25], were mycoplasma tested several times per year, and were validated by endoglycosidase H treatment of secreted glycoprotein. Briefly, residues 1–594 of the $\alpha V$-subunit and 1–456 of the $\beta 8$-subunit were cloned into modified pcDNA3.1 and ET10 expression vectors, respectively. Mutations M400C in $\alpha V$ and V259C in $\beta 8$ formed a disulfide to covalently stabilize the heterodimer. Purification with Ni-affinity chromatography, removal of tags, ion exchange, and gel filtration were as described for $\alpha V\beta 6$[17], except the gradient with Sepharose Q was from 50 to 150 mM NaCl and gel filtration was with Superdex 75 in 20 mM HEPES, pH 7.5, 150 mM NaCl, 1 mM $MgCl_2$, and 1 mM $CaCl_2$. Fractions were concentrated to 4.5 mg/ml and stored at −80 °C in aliquots. Mutant fragments in the same expression vector were transiently expressed with the $\alpha V$-subunit using FetcoPro (PolyPlus, Strasbourg, France) in suspension Expi293 cells. Supernatants were collected after 6 days and protein was purified as described above. Human pro-TGF$\beta$-1 with a R249A cleavage site mutation was prepared as described[20].

**Crystal structures**. Hanging drop $\alpha V\beta 8$ headpiece crystals grew in 0.1 M MES (2-($N$-morpholino)ethanesulfonic acid), pH 6.7, 12% polyethylene glycol (PEG)

20,000 at 4 °C. To improve electron density on the β8-subunit, crystals were dehydrated by soaking in solutions that had the starting concentrations of components in the protein and reservoir solutions while raising the concentration of PEG 20,000 to 20% in 2% steps. Additionally, soaking solutions contained 20 mM $Mg^{2+}$ and 10 mM $Ca^{2+}$ (unliganded structure) or 1 mM TGF-β1 ligand peptide ($G^{213}$RRGDLATIHG$^{223}$), 10 mM $Mg^{2+}$, and 2 mM $Ca^{2+}$ (liganded structure). Processing and diffraction limit determination were with XDS[26] and $CC_{1/2}$[27], respectively. The liganded structure was solved by molecular replacement with αVβ6 (4UM9) using PHASER in Phenix[28] and subsequently used to solve the unliganded structure. Structures were refined with PHENIX, built with Coot[29], and validated with MolProbity[30]. Representative electron density is shown in Supplementary Fig. 6. Figures were made with PyMol (Schrödinger, NY, NY). Structural data have been deposited in the Protein Data Bank under accession numbers 6OM1 for unliganded αVβ8 and 6OM2 for liganded αVβ8.

**Fluorescence polarization**. Saturation binding was measured in HBS buffer (20 mM HEPEs, pH 7.5, 150 mM NaCl), supplemented with 1 mM $Mg^{2+}$/1 mM $Ca^{2+}$ or 1 mM $Mn^{2+}$/0.2 mM $Ca^{2+}$ with fluorescein isothiocyanate (FITC)-labeled pro-TGF-β3 peptide (FITC-Aminocaproic-GRGDLGRLKK) probe. αVβ8 was serially diluted in 1.4-fold decrements and mixed with 5 nM of probe at 20 °C for 30 min. Fitting fluorescence polarization as a function of integrin concentration at fixed probe concentrations yielded $K_D$ values for fluorescent pro-TGF-β3 peptide[14].

**Electron microscopy**. αVβ8-mut3 or αVβ8-mut6 (15 μg) were mixed with pro-TGF-β1 at molar ratio 1.5:1 in 50 μl of HBS buffer containing 1 mM $Mn^{2+}$/0.2 mM $Ca^{2+}$ for 30 min and injected in a 24 ml Superdex 75 gel filtration column pre-equilibrated with HBS buffer (20 mM HEPEs, pH 7.5, 150 mM NaCl, 1 mM $Mn^{2+}$/0.2 mM $Ca^{2+}$). The 2:2, 2:1 complexes and unbound αVβ8 and pro-TGF-β1 were well separated[14]. Peak complex fractions (~5 μg/ml, as estimated by $A_{280}$) were loaded on glow-discharged carbon grids and fixed with uranyl formate. About 60 images with 52,000 magnification were collected on FEI Tecnai-12 transmission electron microscope at 120 kV with low-dose model. About 5000 particles were manually picked and subjected to multireference alignment and $K$-means classification by software SAMUEL[31].

**Hydrogen-deuterium exchange**. Two microliters of αVβ6 and αVβ8 stock solutions (23 and 33 μM, respectively), either alone or mixed with 100 μM TGF-β1 ligand peptide, were diluted 15-fold (to 1.53 and 2.20 μM, αVβ6 and αVβ8, respectively, and 6.67 μM TGF-β1 ligand peptide) with labeling buffer (Supplementary Table 1) at 21 °C to initiate deuterium exchange. Peptide $K_D$ values of 12 and 30 nM for αVβ6 and αVβ8 (Fig. 7), respectively, predict binding to 99.8% and 99.6% of the integrin, respectively. At time points from 10 s to 240 m, an aliquot was removed and an equal volume of quench buffer (Supplementary Table 1) was added to adjust the pH to 2.5. Each sample (46 pmol αVβ6 or 66 pmol αVβ8) was immediately subjected to liquid chromatography-mass spectrometry analysis as described in the next paragraphs and in our PRIDE submission (PXD014348). For binding experiments, protein and ligand were allowed to equilibrate for 20 min at 21 °C before deuterium labeling.

Deuterated and control samples were digested in solution with pepsin (10 mg/ml, Sigma P6887; Lot#SLBL1721V) for 5 min on ice, and then injected into an M-class Acquity UPLC with HDX technology (Waters)[32]. The cooling chamber of the UPLC system, which housed all the chromatographic elements was held at 0.0 ± 0.1 °C for the entire time of the measurements. The injected peptides were trapped and desalted for 3 min at 100 μl/min using a BEH C18 2.1 × 5 mm² column (Waters, 186003975) and then separated in 14 min by a 5–40% acetonitrile:water gradient at 40 μl/min. The separation column was a 1.0 × 100.0 mm² Acquity UPLC C18 BEH (Waters, 186002346) containing 1.7 μm particles. The back pressure averaged 8800 psi at 0.1 °C[33]. The error of determining the deuterium levels was ±0.15 Da in this experimental setup. To eliminate peptide carryover, a wash solution of 1.5 M GnHCl, 0.8% formic acid, and 4% acetonitrile was injected after each run.

Mass spectra were acquired using a Waters Synapt G2-Si HDMS$^E$ mass spectrometer in ion mobility mode. A conventional electrospray source was used and the instrument was scanned over the range 100 to 1900 $m/z$. The instrument configuration was the following: capillary was 3.2 kV, trap collision energy at 6 V, sampling cone at 35 V, source temperature of 80 °C, and desolvation temperature of 175 °C. All comparison experiments were done under identical experimental conditions such that deuterium levels were not corrected for back-exchange and are therefore reported as relative[34].

Peptides were identified using PLGS 3.0.1 (Waters, 720001408EN) using six replicates of undeuterated αVβ6 and six replicates of undeuterated αVβ8. Raw data were imported into DynamX 3.0 (Waters, 720005145EN) and filtered as follows: minimum number of products of 3; minimum consecutive products of 2; minimum number of products per amino acid of 0.2; maximum mass error of 10 p.p.m. Those peptides meeting the filtering criteria were further processed automatically by DynamX followed by manual inspection of all processing. Peptides with low signal-to-noise ratios in either bound or free states were removed. The relative amount of deuterium in each peptide was determined by subtracting the centroid mass of the undeuterated form of each peptide from the deuterated form, at each time point, for each condition. These deuterium uptake

values were used to generate uptake graphs and difference maps. Additional experimental details are found in Supplementary Table 1.

**Reporting summary**. Further information on research design is available in the Nature Research Reporting Summary linked to this article.

## Data availability

Raw HDX MS data have been have been deposited to the ProteomeXchange Consortium via the PRIDE partner repository[35] with the data set identifier PXD014348 [http://proteomecentral.proteomexchange.org/cgi/GetDataset?ID=PXD014348]. Source data and fit values have been provided for Fig. 7a, b and Supplementary Fig. 3a–t. Each panel's data appears as a tab in an excel file. The fit values reported in Fig. 7a come from each tab in this file. Figure 7b is a representative of one of the panels in Supplementary Fig. 3. Protein database accession IDs are 6OM1 for unliganded αVβ8 headpiece and 6OM2 for liganded αVβ8 headpiece. All other data are available from the corresponding author on reasonable request.

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

## Acknowledgements

This work was supported by NIH grant HL-134723 and a research collaboration with Waters Corporation (J.R.E.). Diffraction data were acquired at GM/CA beamline 23-ID of the Advanced Photon Source at Argonne National Laboratory. We would like to thank Prof. Thomas Wales for useful discussion and Ms. Margaret Nielsen for Illustrator assistance.

## Author contributions

J.R.E. and T.A.S. supervised the work, analyzed data, and wrote the paper. J.W. and R.E.I. performed experiments, analyzed data, and wrote the paper. Y.S. developed scripts for analyzing and graphically displaying HDX data and prepared figures.

## Competing interests

T.A.S. is a stock owner, consultant, and board member of Morphic Therapeutic. Other authors declare no competing interest.
