## [Peer Review File · Nature Communications]

Reviewers' comments:

Reviewer #1 (Remarks to the Author):

This manuscript is of great interest because it provides new insights into integrin signaling. Amongst many important functions in cells, integrin mediates the activation of the transforming growth factor beta (TGFbeta) which effects stromal processes and the integrin alphaVbeta8-mediated TGFbeta activation by effector regulatory T cells is crucial for the suppression of T-cell mediated inflammation. Integrin alphaV also regulates the TGFbeta promotion of cancer cells.

Wang and colleagues determined the crystal structures of the headpiece of integrin alphaVbeta8 on its own and in complex with the TGFbeta1. These structures are of particular interest because of the 24 integrins, integrin alphaVbeta8 couples to the actin cytoskeleton by a different mechanism. While most integrins link extracellular ligands to the actin cytoskeleton by cellular traction via adaptor proteins like talin, integrin alphaVbeta8 binds to the band 4.1 family instead. In the classic integrin activation mechanism, the extended-open integrin conformer is stabilized by force and ligand binding whereas the integrin alphaVbeta8 ectodomain is in its constitutively extended form without activation by force.

The structures from the Springer laboratory presented here provide important details and uncover new and unexpected regions of the I-domains of talin-activated integrins versus the I-domain of integrin beta8 (this manuscript) outside the adjacent metal ion dependent adhesion site (AMIDAS). The authors confirm their structural findings by mutagenesis, hydrogen-deuterium exchange, binding studies, and negative stain electron microscopy of integrin alphaVbeta8 as well as alphaVbeta6 (that shares the distinct integrin coupling mechanism and the TGFbeta binding ligand).

Some minor suggestions:

1. The title (and keywords) is perhaps too modest, how about something like:

Atypical activation mechanism of integrin alphaVbeta8 by high affinity binding of the transforming growth factor beta-1

2. The first paragraph of the introduction is perhaps also too modest, and it would perhaps be of interest to the general audience to provide some more background on the importance of integrins in controlling so many key events in a cell as well as their roles in diseases

3. While it's perfectly acceptable to cite their 2017 PNAS paper with respect to the constructs used here, it would be helpful to clarify on page 3 that they used M400C mutant alphaV residues 1-594 and V259C mutant beta8 residues 1-456 and/or at least in the methods section please and please remind the general audience about the rationale of using these mutations

4. Page 3, please quantify the statement that the electron density is weaker for the hybrid domain for example by providing temperature factor information and please state (in text and/or Figure legend) what residues were not built in the plexin-semaphorin-integrin (PSI) and integrin epidermal growth factor-like (I-EGF) domains; similarly, on page 6, second last paragraph

5. So D8 and D9 in the text (and Figure 3G) are N119 and N120 in Figure 3A and 3B; maybe it would help to explicitly state this somewhere especially since the text also uses Asn-120 later on (perhaps it would be clearer to not renumber the residues in the text or, less preferred, to also renumber them in the Figure)

6. Page 5 and/or in the methods, please add residue numbers to the peptide 242-GRRGDLATIHG-252

7. Figure 2G is particularly helpful showing a clear and distinct alpha1 helix in the open, unliganded integrin alphaIIbBeta3 structure while also highlighting the structural differences of the alpha7 helix in the open, unliganded integrin alphaIIbBeta3 structure and the closed, unliganded alphaIIbBeta3 structures compared to the other three structures

8. Pages 5 and 6, maybe it would help the general audience to add the mentioned states to Figure 1

9. I am wondering what effects crystal contacts have and if any of the regions discussed is are in contact with a symmetry-related molecule

10. Figure 3G, what is the sequence similarity and identity over the entire polypeptide chains (both alpha, V and IIb in particular, and beta)?
11. Page 7, maybe it would help to define the +2 and -1 positions to the general audience
12. I am curious whether integrin beta8 residues Asn-119, Asn-120, and Gln-302 are conserved in other species
13. Figure 6 might be better as a Supplementary Figure as it otherwise perhaps dilutes the message of the data Figures
14. It seems that this team determined the crystal structures with and without dehydration so it would be interesting to know, even at low resolution, if there were any major differences seen in the dehydrated versus hydrated crystal structures
15. Please provide the protein concentrations used for electron microscopy (are they 12 nM and 8 nM, as for the gel filtration as stated in the supplement?) in Figure 5c and in the methods section
16. Figure 1, perhaps line up the top line on the far right with the top of the schematic? And please label the specificity-determining loop 1(SDL1) mentioned in the legend and please point the beta1 label to its schematic
17. Apologies if I missed it, but how many repeats were there for the HDX?
18. Figure 5C, what was the pro-TGF-b1 construct, was it full-length? Please provide residue numbers for all proteins used in each experiment
19. Perhaps it would be clearer to replace “D incorporation for bound integrin – D incorporation for unbound integrin” with something like “differential deuterium incorporation of bound versus unbound integrin” on pages 24 and 25

20. Perhaps it would be better to have Supplemental Figure S3 in portrait instead of the current landscape and to label each panel (A through T) with a legend
21. Supplemental Figure S4, if you have a native and an SDS-PAGE gel (and Western blot) of the fractions a through d, please provide these. And please mention in the legend what the calculated and apparent molecular weights are for the peaks a through d.
22. Supplemental Figure S5b is a very elegant way of providing a visual instead of the many words it would take to convey that message
23. Supplemental Table 1 might be better in the main text (please provide units for temperature factors, line up the values with the clash/geometry instead, and please specify somewhere in table that “unliganded” is the “unliganded integrin M400C mutant alphaV residues 1-594 and V259C mutant beta8 residues 1-456). The X-ray diffraction data seem noisy, which is probably simply the limitation of the crystals, with low overall $I/\sigma(I)$ and low values in the last shells. Perhaps lowering the resolution limit might help. The two datasets are also low on symmetry with low redundancy especially for the unliganded structure. Did the authors choose an X-ray data collection protocol specifically for the low symmetry of the crystals and if so, what were the difficulties? Again, revising the resolution downward might improve the statistics. Please provide a plot of $I/\sigma(I)$ in resolution shells for the reviewers. Finally, how was the NCS used during the refinement?

Reviewer #2 (Remarks to the Author):

The manuscript by Wang et al investigates the atypical activation mechanism of integrin $\alpha V\beta 8$ through a combination of approaches including solving the x-ray in the presence and absence of pro-TGF- $\beta 1$ peptide, comparisons of the conformational dynamics of $\alpha V\beta 8$ and $\alpha V\beta 6$ w/o pro-TGF- $\beta 1$ peptide, and affinity measurements of mutants. The structure and mutagenesis results on $\alpha V\beta 8$

indicate that the $\beta 8 \beta 1$ domain have atypical structure features that may allow it to “open” in the absence of the larger scale movement of the hybrid domain (associated with an extended-open state) seen in other integrins. As there were disordered regions in the $\beta 8$ that was not resolved in the crystal structure, it is important that this work is supported by EM - and HDX-MS data that probe native solution-phase dynamics. HDX-MS was used to compare backbone dynamics of $\alpha V\beta 8$ and $\alpha V\beta 6$. The data indicate that while $\alpha V\beta 8$ and $\alpha V\beta 6$ have very similar dynamics in the αV subunits, interesting differences HDX in the $\beta 1$ domain and SDL2 may occur (in apparent support of the

disorder observed in the crystal structure), however a more thorough treatment of the data and control experiments are needed to conclude on this. The response to ligand binding in the β I domain (α 1 region) further shows interesting differences between α V β 8 and α V β 6 supporting the conclusions drawn from the X-ray data. Overall, I find the work interesting and the results from the combination of techniques used very convincing. I am positive towards publication provided the below comments are addressed.

Comments:

The authors should discuss how the absence of other parts of the ectodomain could impact the reported dynamics and conformational changes upon ligand binding of β 8/ β 6. Did the authors do similar work with the entire ectodomain?

“Multiple peptides covering the β I domain α 1 and α 1' helices showed more rapid exchange of backbone amide hydrogens in β 8 than β 6 (Fig. 4A-D and Supplemental Figs. 1 and 2)”. The authors must be careful making such comparisons if substantial sequence differences exist between β 8 and β 6– the impact on kch will impact both in-exchange and back-exchange. Please clarify if appropriate controls were performed to account for this (kch calculations, analysis of a maximally-labeled sample etc.).

The authors should justify their use of a 1Da cut-off for “meaningful” differences in HDX upon peptide binding to β 8 and β 6. It may be justified but it looks to me like they could be under-analyzing their data since they have triplicates measurements and could do a more well-defined statistical treatment of the data etc?

Figure 4f: The authors need to discuss the Δ HDX differences between β 8 than β 6 in more detail. Even with their very conservative threshold for “meaningful” HDX changes there seems to be other regions that are differently impacted by peptide binding including SDL2-SDL3 (β 3 region)

Figure 4ef: Curves for each timepoint are overlaid in such small figures making it hard to follow the effects at the earlier timepoints. I recommend the authors to have the difference plots showing curves for the earliest timepoints in front (i.e. red, orange) – as these timepoint are more likely to reflect changes upon ligand binding (all other things being equal).

“Deuterated and control samples were digested in solution with pepsin (10 mg/mL) for 5 minutes on ice, then injected into a custom Waters UPLC HDX Manager”. The choice of doing in-solution digest seems slightly odd to me – but this could be related to improved reduction? Could the authors comment on this?

The authors are commended for including the SI HDX Summary and HDX Data tables in the supporting information as recommended in the community white paper that is due to be published in Nature Methods in July. The authors are encouraged to make a reference in the methods section to this community-based white paper that outlines the format and details the value of including these data tables.

Reviewer #3 (Remarks to the Author):

The authors present crystallographic, hydrogen deuterium exchange dynamics, and negative stain EM data to dissect the atypical features of $\alpha V\beta 8$ in proTGF- $\beta 1$ ligand binding. They report that: 1) $\beta 8$ lacks an ADMIDAS ion, 2) the $\alpha 1$ and $\alpha 1'$ helices in the βI domain have more rapid deuterium exchange dynamics, 3) the $\beta 6$ - $\alpha 7$ loop structure in $\beta 8$ has a unique conformation and unlike other integrin receptors studied to date does not engage the $\alpha 1$ helix, and 4) the SDL1 moves toward the MIDAS with ligand binding, assuming an intermediate position between those of the bent-closed and extended-open positions in $\alpha II\beta 3$. A series of mutant $\alpha V\beta 8/\alpha V\beta 6$ receptors elucidate some of the key interactions that account for these findings. The discovery of the conserved structures of residue that stabilize the turn between the $\beta 6$ -strand and $\alpha 7$ -helix in many β subunits but not $\beta 8$ is particularly notable because it helps ligate the $\beta 6$ - $\alpha 7$ loop to the ADMIDAS metal ion.

The experimental design is comprehensive and benefits from the use of several complementary structural and functional methods. The manuscript is precisely written, but the complexity and density of the information demands undivided attention by the reader.

1. In the Introduction, it is stated that the $\alpha V\beta 8$ ectodomain is extended, but this statement does not have a reference. The first paper to report this important finding should be cited.

2. The authors should discuss whether any of their findings, in particular, the lack of an interaction between the $\beta 6$ - $\alpha 7$ loop and the ADMIDAS Ca^{2+} , is responsible for the receptor assuming an extended conformation.

3. The authors point to the interaction of N120 with the $\beta 8$ -unique Q302 as potentially contributing to the loss of the ADMIDAS metal ion. Did the authors make a mutant in which just the Q302 was replaced with a T302? Mutant 1 in their series had mutations of both N120D and Q302T.

4. Ligand binding section on page 5. Since the numbering of the ligand residues overlaps the numbering of the $\alpha V\beta 3$ residues, it would be good to indicate the numbering at the beginning and end of the RGDLATI sequence. Similarly, figure 2 would be easier to interpret if the ligand residue numbers were in the same color as the ligand itself to distinguish them from the $\alpha V\beta 8$ sequence residues.

5. The authors emphasize the importance of the $\beta 8$ S116 directly interacting with the MIDAS metal ion in the liganded $\alpha V\beta 8$ receptor adopting the high affinity conformation. At the same time, they report that S127 in $\alpha V\beta 6$ does not directly engage the MIDAS metal ion in liganded $\alpha V\beta 6$, but do not comment on the significance of that finding. This would benefit from additional discussion.

6. At the end of the Results section, this reviewer was surprised that the interesting $\alpha V\beta 6$ mutants in Figure 5A were not discussed. These mutants are later discussed at length in the Discussion. The authors should consider reporting the $\alpha V\beta 6$ mutants data in the Results section and then focusing just on the implications in the Discussion section.

7. Page 6. To facilitate the interpretation of Figure 3A-F, text should be modified to “the vertical dashed lines in Fig. 3A-F mark the position of the state 1 $\alpha \text{IIb}\beta 3$ D126 carbon atom sphere.”

8. Page 6. The authors should explain why movement of the S5 residue to the MIDAS should increase affinity for ligand.

9. Page 6, last line. It appears that the authors are referring to Fig. 3G rather than 2G.

10. Page 8. The authors note that ligand binding enhances α 1 helix hydrogen deuterium exchange, but in figure 4 it appears that exchange in α 1 helix varies over time after ligand binding, with only some time points showing enhancement.

11. Page 14. The authors refer to reference 18 (Hu and Luo) which showed that N8/N9 α V β 3 has similar vitronectin binding to that of WT α V β 3 whereas D8/D9 α V β 8 has decreased binding of vitronectin relative to WT α V β 8. They acknowledge the difference with the enhanced RGD/LATI ligand binding they observed with mut8 (D8/D9 α V β 8), but do not provide any explanation. They allude to “vitronectin binding, rather than affinity,” but the meaning of this is not clear to this reviewer. Since Hu and Luo reported that the leg regions of α V β 8 play a role in receptor ligand binding affinity, the authors should discuss the impact of their studying the α V β 8 headpiece with an engineered disulfide bond rather than the intact α V β 8 receptor.

12. Page 14. The authors refer to the ADMIDAS metal ion coordinating the Asp in the β 4- α 5 loop and stabilizing its position in the outer coordinate shell of the MIDAS metal ion. The authors should consider putting the sequence of this loop in Figure 3G. What is the evidence for the interaction being in the outer shell?

13. Page 16. The description of the Fab that permits the β I to be open when the hybrid domain is either closed or open should be described in more detail.

Dear Reviewers,

You Reviewers were unusually perceptive and we have improved the MS in response. We have expanded on the MS to improve its comprehension and better refined our arguments for deductions and soften some of the HDX interpretations.

The last figure was moved up to Fig. 3 to provide more perspective on conformational changes in integrins and more HDX data has been moved to main text and the HDX figure was split into two figures. We believe that this has improved the MS and are proud to resubmit it. We show our responses below in blue text.

Because there is a competing CryoEM structure being submitted elsewhere we request that reviews be as expeditious as possible. And apologize that this is coming in August.

Reviewer #1 (Remarks to the Author):

This manuscript is of great interest because it provides new insights into integrin signaling. Amongst many important functions in cells, integrin mediates the activation of the transforming growth factor beta (TFGbeta) which effects stromal processes and the integrin alphaVbeta8-mediated TFGbeta activation by effector regulatory T cells is crucial for the suppression of T-cell mediated inflammation. Integrin alphaV also regulates the TFGbeta promotion of cancer cells.

Wang and colleagues determined the crystal structures of the headpiece of integrin alphaVbeta8 on its own and in complex with the TFGbeta1. These structures are of particular interest because of the 24 integrins, integrin alphaVbeta8 couples to the actin cytoskeleton by a different mechanism. While most integrins link extracellular ligands to the actin cytoskeleton by cellular traction via adaptor proteins like talin, integrin alphaVbeta8 binds to the band 4.1 family instead. In the classic integrin activation mechanism, the extended-open integrin conformer is stabilized by force and ligand binding whereas the integrin alphaVbeta8 ectodomain is in its constitutively extended form without activation by force.

The structures from the Springer laboratory presented here provide important details and uncover new and unexpected regions of the I-domains of talin-activated integrins versus the I-domain of integrin beta8 (this manuscript) outside the adjacent metal ion dependent adhesion site (AMIDAS). The authors confirm their structural findings by mutagenesis, hydrogen-deuterium exchange, binding studies, and negative stain electron microscopy of integrin alphaVbeta8 as well as alphaVbeta6 (that shares the distinct integrin coupling mechanism and the TGFbeta binding ligand).

Thanks for your positive comments!

Some minor suggestions:

1. The title (and keywords) is perhaps too modest, how about something like: "Atypical activation mechanism of integrin alphaVbeta8 by high affinity binding of the transforming growth factor beta-1"

It is refreshing to be called modest. We have revised title to:

"Structurally atypical activation mechanism of integrin $\alpha V\beta 8$ "

2. The first paragraph of the introduction is perhaps also too modest, and it would perhaps be of interest to the general audience to provide some more background on the importance of integrins in controlling so many key events in a cell as well as their roles in diseases

We have added more context by expanding the beginning of the Introduction.

3. While it's perfectly acceptable to cite their 2017 PNAS paper with respect to the constructs used here, it would be helpful to clarify on page 3 that they used M400C mutant alphaV residues 1-594 and V259C mutant beta8 residues 1-456 and/or at least in the methods section please and please remind the general audience about the rationale of using these mutations.

We added a sentence to Methods to call this out.

4. Page 3, please quantify the statement that the electron density is weaker for the hybrid domain for example by providing temperature factor information and please state (in text and/or Figure legend) what residues were not built in the plexin-semaphorin-integrin (PSI) and integrin epidermal growth factor-like (I-EGF) domains; similarly, on page 6, second last paragraph.

It is hard to compare B factors among different structures that differ in data quality, resolution, etc.

Therefore, we have quantitated the number of residues that were built in results on page 3: “Among all independent unliganded and liganded structures, respectively, the average number of residues that could be built was 64 and 74% ($\beta 6$) and 0 and 0% ($\beta 8$) for PSI, 100 and 100% ($\beta 6$) and 71 and 61% ($\beta 8$) for hybrid, 100 and 100% ($\beta 6$) and 96 and 100% ($\beta 8$) for $\beta 1$, and 11 and 81% ($\beta 6$) and 0 and 0% ($\beta 8$) for I-EGF1”. On page 6, the position of the missing residues was already defined, as between the a1 and a1' helices, and is shown in a Figure.

5. So D8 and D9 in the text (and Figure 3G) are N119 and N120 in Figure 3A and 3B; maybe it would help to explicitly state this somewhere especially since the text also uses Asn-120 later on (perhaps it would be clearer to not renumber the residues in the text or, less preferred, to also renumber them in the Figure)

We say:

“For ease of nomenclature here, we define the contiguous sequence of MIDAS and ADMIDAS-coordinating residues in typical integrin β -subunits, DXSXSXXDD (D1-S3-S5-D8-D9), as the β -MIDAS motif, where MIDAS is used in a broad sense to include up to two metal ions (Fig. 3G). In typical integrins, the sidechains of the two Asp residues (β -MIDAS D8 and D9), the backbone carbonyl of the β -MIDAS S5 residue, and a backbone carbonyl from the $\beta 6$ - $\alpha 7$ loop coordinate the ADMIDAS metal ion (Fig. 3C-G). In contrast, $\beta 8$ has Asn-119 and Asn-120 (N8 and N9) in place of the D8 and D9 Asp residues (Fig. 3A, B, and G).”

6. Page 5 and/or in the methods, please add residue numbers to the peptide 242-GRRGDLATIHG-252

So done.

7. Figure 2G is particularly helpful showing a clear and distinct alpha1 helix in the open, unliganded integrin alphaIIbBeta3 structure while also highlighting the structural differences of the alpha7 helix in the open, unliganded integrin alphaIIbBeta3 structure and the closed, unliganded alphaIIbBeta3 structures compared to the other three structures

Thank you.

8. Pages 5 and 6, maybe it would help the general audience to add the mentioned states to Figure 1.

These movements concern small portions of the b1 domain only and would be very difficult to get to show up in a small cartoon.

9. I am wondering what effects crystal contacts have and if any of the regions discussed is are in contact with a symmetry-related molecule

Good question. We have added the sentence “None of these differences are at lattice contacts in the $\alpha V \beta 8$ crystal structures.”

10. Figure 3G, what is the sequence similarity and identity over the entire polypeptide chains (both alpha, V and IIb in particular, and beta)?

We now state in the text that: “Over the entire ectodomain, $\beta 8$ and $\beta 6$ are 40% identical, and identity is highest in the $\beta 1$ domain, at 48%.” The aV and aIIb subunits are in the same RGD-binding subfamily of integrin a subunits and are 38% identical. Identity of b8 to other beta subunits varies from 34 to 40%.

11. Page 7, maybe it would help to define the +2 and -1 positions to the general audience

Yes, we now do.

12. I am curious whether integrin beta8 residues Asn-119, Asn-120, and Gln-302 are conserved in other species

Good question. To Discussion, first paragraph, we have added:

“We examined conservation of N8 (Asn-119), N9 (Asn-120), and Gln-302 in integrin $\beta 8$ in evolution. All three residues are invariant in mammals and chicken. Among fish (zebrafish, Japanese rice fish, spotted gar, and elephant shark), only Asn-120 (N9) is invariant. Asn-119 (N8) is found as Asp (twice), Glu, and Ala. Gln-302 is found as Asp (twice), Glu, and Gln. The sidechains of all of these residues at position 302 would be capable of hydrogen bonding to the invariant Asn-120 residue at position N9”

13. Figure 6 might be better as a Supplementary Figure as it otherwise perhaps dilutes the message of the data Figures

We believe it is interesting and it does contain data on how much our aVb8 structures have opened. The sigmoid shape of this curve is interesting and has not been pointed out before. We have now made this Figure 3 of Results and used it more to provide context on aVb8.

14. It seems that this team determined the crystal structures with and without dehydration so it would be interesting to know, even at low resolution, if there were any major differences seen in the dehydrated versus hydrated crystal structures

Correct. But in non-dehydrated structures, density of the bl domain was poor, with 30% of residues disordered, including some of the critical ones discussed here. Dehydration significantly improves map quality. Dehydration also changed space group and cell parameters. The space group of hydrated structure was P212121, and axis parameters were a=191.4 Å, b=54.9 Å and c= 290.6 Å.

15. Please provide the protein concentrations used for electron microscopy (are they 12 nM and 8 nM, as for the gel filtration as stated in the supplement?) in Figure 5c and in the methods section

“Peak complex fractions (~5 µg/ml, as estimated by A_{280}) were loaded on glow-discharged carbon grids and fixed with uranyl formate.”

16. Figure 1, perhaps line up the top line on the far right with the top of the schematic? And please label the specificity-determining loop 1(SDL1) mentioned in the legend and please point the beta1 label to its schematic.

So done.

17. Apologies if I missed it, but how many repeats were there for the HDX?

There were three, and this was described in Supplemental Table S2 that describes all the HDX MS experimental parameters.

18. Figure 5C, what was the pro-TGF-b1 construct, was it full-length? Please provide residue numbers for all proteins used in each experiment

Yes. Now in Methods:

“Full-length human pro-TGF β -1 with C4S mutation to remove the cysteine that links to a milieu molecule and R249A furin cleavage site mutation (mature numbering) was prepared as described.”

19. Perhaps it would be clearer to replace “D incorporation for bound integrin – D incorporation for unbound integrin” with something like “differential deuterium incorporation of bound versus unbound integrin” on pages 24 and 25

It is important to describe the equation for subtraction in a differential HDX experiment so it is clear what was subtracted from what, so one knows how to interpret the sign (positive or negative as a result of what). In our case, unbound D level was subtracted from bound D level so negative values mean protection upon binding. We have changed the Fig 4 legend to “the equation for subtraction was ($D_{\text{liganded}} - D_{\text{unliganded}}$)”.

20. Perhaps it would be better to have Supplemental Figure S3 in portrait instead of the current landscape and to label each panel (A through T) with a legend.

We have done so.

21. Supplemental Figure S4, if you have a native and an SDS-PAGE gel (and Western blot) of the fractions a through d, please provide these. And please mention in the legend what the calculated and apparent molecular weights are for the peaks a through d.

We don't have these. Please see our previous paper. There we compared profiles from injecting pro-TGF- β 1 or integrin alone, or the mixture of them. This provided strong evidence for complex formation and identity of these peaks. We now cite this in the figure legend. "The identity of labeled peaks a-d was previously established (Wang et al, PNAS, 2017)."

22. Supplemental Figure S5b is a very elegant way of providing a visual instead of the many words it would take to convey that message.

23. Supplemental Table 1 might be better in the main text (please provide units for temperature factors, line up the values with the clash/geometry instead, and please specify somewhere in table that "unliganded" is the "unliganded integrin M400C mutant alphaV residues 1-594 and V259C mutant beta8 residues 1-456). The X-ray diffraction data seem noisy, which is probably simply the limitation of the crystals, with low overall $I/\sigma(I)$ and low values in the last shells. Perhaps lowering the resolution limit might help. The two datasets are also low on symmetry with low redundancy especially for the unliganded structure. Did the authors choose an X-ray data collection protocol specifically for the low symmetry of the crystals and if so, what were the difficulties? Again, revising the resolution downward might improve the statistics. Please provide a plot of $I/\sigma(I)$ in resolution shells for the reviewers. Finally, how was the NCS used during the refinement?

We make this Main Table 1. We have made the indicated changes. The low overall $I/\sigma(I)$ and low values in the last shells are typical when choosing the cutoff with CC 1/2. Our object is to include all the data that helps to obtain a better model rather than to improve statistics. The plot is included.

Indeed, we used NCS during refinement. We then removed NCS during final refinement rounds. IN response to the question, we checked identical refinements with and without NCS. The Rwork and Rfree differed only by 1 in the last (fourth) significant figure.

Reviewer #2 (Remarks to the Author):

The manuscript by Wang et al investigates the atypical activation mechanism of integrin $\alpha V\beta 8$ through a combination of approaches including solving the x-ray in the presence and absence of pro-TGF- $\beta 1$ peptide, comparisons of the conformational dynamics of $\alpha V\beta 8$ and $\alpha V\beta 6$ w/o pro-TGF- $\beta 1$ peptide, and affinity measurements of mutants. The structure and mutagenesis results on $\alpha V\beta 8$ indicate that the $\beta 8$ $\beta 1$ domain have atypical structure features that may allow it to “open” in the absence of the larger scale movement of the hybrid domain (associated with an extended-open state) seen in other integrins. As there were disordered regions in the $\beta 8$ that was not resolved in the crystal structure, it is important that this work is supported by EM - and HDX-MS data that probe native solution-phase dynamics. HDX-MS was used to compare backbone dynamics of $\alpha V\beta 8$ and $\alpha V\beta 6$. The data indicate that while $\alpha V\beta 8$ and $\alpha V\beta 6$ have very similar dynamics in the αV subunits, interesting differences HDX in the $\beta 1$ domain and SDL2 may occur (in apparent support of the disorder observed in the crystal structure), however a more thorough treatment of the data and control experiments are needed to conclude on this. The response to ligand binding in the $\beta 1$ domain ($\alpha 1$ region) further shows interesting differences between $\alpha V\beta 8$ and $\alpha V\beta 6$ supporting the conclusions drawn from the X-ray data. Overall, I find the work interesting and the results from the combination of techniques used very convincing. I am positive towards publication provided the below comments are addressed.

Comments:

The authors should discuss how the absence of other parts of the ectodomain could impact the reported dynamics and conformational changes upon ligand binding of $\beta 8/\beta 6$. Did the authors do similar work with the entire ectodomain?

We did not. However, we did for $\alpha 5\beta 1$ (unpublished) and there were no differences in headpiece dynamics between the two. We have softened conclusions in the Results HDX section.

“Multiple peptides covering the $\beta 1$ domain $\alpha 1$ and $\alpha 1'$ helices showed more rapid exchange of backbone amide hydrogens in $\beta 8$ than $\beta 6$ (Fig. 4A-D and Supplemental Figs. 1 and 2)”. The authors must be careful making such comparisons if substantial sequence differences exist between $\beta 8$ and $\beta 6$ – the impact on k_{ch} will impact both in-exchange and back-exchange. Please clarify if appropriate controls were performed to account for this (k_{ch} calculations, analysis of a maximally-labeled sample etc.).

As the reviewer points out, we must be careful in such interpretations. One cannot directly compare exchange between proteins with greatly different sequence without considering the back-exchange of a totally deuterated control or without calculating and considering the intrinsic rates of exchange (k_{ch}). One is able to compare the effects of binding within a single protein without regard to sequence differences. A totally deuterated control could not be prepared for these proteins.

Having calculated the k_{ch} for back-exchange – the key point to consider – we find that one finds an issue in comparisons for B6 and B8 approximately 15.2% of the residues [based on the criteria we published previously when describing this phenomenon and how to handle it (Wales et al, 2016, JASMS, 27, 1048)] across the portion of the proteins that are most conserved (residues Y114-L363 of B6 and Y103-I342 of B8). We have revised all text that makes comparisons between HDX of B6 and B8 to include the word “generally” and explain why one cannot compare directly and accurately quantitate in the first instance of such statements of the revision.

The authors should justify their use of a 1Da cut-off for “meaningful” differences in HDX upon peptide binding to $\beta 8$ and $\beta 6$. It may be justified but it looks to me like they could be under-analyzing their data since they have triplicates measurements and could do a more well-defined statistical treatment of the data etc?

We have added two sentences to the HDX section of Results:

“We chose a 1 Da cutoff to mark HDX differences that are clearly above noise and are likely meaningful structurally. Other changes in the range of 0.5-1.0 Da are above triplicate variation and may also have limited importance.”

Figure 4f: The authors need to discuss the deltaHDX differences between $\beta 8$ than $\beta 6$ in more detail. Even with their very conservative threshold for “meaningful” HDX changes there seems to be other regions that are differently impacted by peptide binding including SDL2-SDL3 ($\beta 3$ region)

To enable fuller understanding of the HDX data, we have changed from one HDX figure to two, with Fig. 4 showing exchange and Fig. 5 showing differences in exchange. To clarify the point the reviewer raises, we added this sentence to the HDX results section:

“Detailed comparisons between $\alpha V\beta 6$ and $\alpha V\beta 8$ are not possible in SDL2 and SDL3 because in addition to the effects of sequence differences on rates of exchange, the lengths of the peptides and the positions of their midpoints plotted in Fig. 5B and D varies, as shown by plotting the peptides (Fig. 5E-H).”

Figure 4ef: Curves for each timepoint are overlaid in such small figures making it hard to follow the effects at the earlier timepoints. I recommend the authors to have the difference plots showing curves for the earliest timepoints in front (i.e. red, orange) – as these timepoint are more likely to reflect changes upon ligand binding (all other things being equal).

This has been done.

“Deuterated and control samples were digested in solution with pepsin (10 mg/mL) for 5 minutes on ice, then injected into a custom Waters UPLC HDX Manager”. The choice of doing in-solution digest seems slightly odd to me – but this could be related to improved reduction? Could the authors comment on this?

We first performed digestion using an online pepsin column; however, the coverage was poor and we found that in-solution digestion provided better coverage.

The authors are commended for including the SI HDX Summary and HDX Data tables in the supporting information as recommended in the community white paper that is due to be published in Nature Methods in July. The authors are encouraged to make a reference in the methods section to this community-based white paper that outlines the format and details the value of including these data tables.

We have added a reference to the HDX MS community standards paper in the methods section.

The HDX Data tables have been removed as they duplicate the PRIDE submission.

Furthermore, we attach information here that will allow the reviewer to log in and check the PRIDE data:

Project accession: PXD014348

Project DOI: Not applicable

Reviewer account details:

Username: reviewer53128@ebi.ac.uk

Password: JNyX16ck

We commend the HDX community for the nice acronym that allows thorough HDX spectroscopists to take PRIDE in their data.

Our data availability statement now reads:

Data availability

Raw HDX MS data have been deposited to the ProteomeXchange Consortium via the PRIDE partner repository³⁸ with the dataset identifier PXD014348. Source data and fit values have been provided for Supplementary Figure 3, panels a-t. Each panel's data appears as a tab in an excel file. The fit values reported in Fig. 7a come from each tab in this file. Fig. 7b is a representative of one of the panels in Supplementary Figure 3.

Reviewer #3 (Remarks to the Author):

The authors present crystallographic, hydrogen deuterium exchange dynamics, and negative stain EM data to dissect the atypical features of $\alpha V\beta 8$ in proTGF- $\beta 1$ ligand binding. They report that: 1) $\beta 8$ lacks an ADMIDAS ion, 2) the $\alpha 1$ and $\alpha 1'$ helices in the βI domain have more rapid deuterium exchange dynamics, 3) the $\beta 6$ - $\alpha 7$ loop structure in $\beta 8$ has a unique conformation and unlike other integrin receptors studied to date does not engage the $\alpha 1$ helix, and 4) the SDL1 moves toward the MIDAS with ligand binding, assuming an intermediate position between those of the bent-closed and extended-open positions in $\alpha IIb\beta 3$. A series of mutant $\alpha V\beta 8/\alpha V\beta 6$ receptors elucidate some of the key interactions that account for these findings. The discovery of the conserved structures of residue that stabilize the turn between the $\beta 6$ -strand and $\alpha 7$ -helix in many β subunits but not $\beta 8$ is particularly notable because it helps ligate the $\beta 6$ - $\alpha 7$ loop to the ADMIDAS metal ion.

The experimental design is comprehensive and benefits from the use of several complementary structural and functional methods. The manuscript is precisely written, but the complexity and density of the information demands undivided attention by the reader.

1. In the Introduction, it is stated that the $\alpha V\beta 8$ ectodomain is extended, but this statement does not have a reference. The first paper to report this important finding should be cited.

This has been done.

2. The authors should discuss whether any of their findings, in particular, the lack of an interaction between the $\beta 6$ - $\alpha 7$ loop and the ADMIDAS Ca^{2+} , is responsible for the receptor assuming an extended conformation.

So done. Sentences have been added to third to last paragraph of Discussion.

3. The authors point to the interaction of N120 with the $\beta 8$ -unique Q302 as potentially contributing to the loss of the ADMIDAS metal ion. Did the authors make a mutant in which just the Q302 was replaced with a T302? Mutant 1 in their series had mutations of both N120D and Q302T.

No, we did not. We mention as possibly contributing. However, it is highly unlikely to be sufficient, since Asn has no negative charge and there are no other negatively-charged residues. Such a \$Ca^{2+}\$ binding site would be unprecedented. Nor would mutation data show whether a \$Ca^{2+}\$ were present or not.

4. Ligand binding section on page 5. Since the numbering of the ligand residues overlaps the numbering of the $\alpha V\beta 3$ residues, it would be good to indicate the numbering at the beginning and end of the RGDLATI sequence. Similarly, figure 2 would be easier to interpret if the ligand residue numbers were in the same color as the ligand itself to distinguish them from the $\alpha V\beta 8$ sequence residues.

This has been done as requested.

5. The authors emphasize the importance of the $\beta 8$ S116 directly interacting with the MIDAS metal ion in the liganded $\alpha V\beta 8$ receptor adopting the high affinity conformation. At the same time, they report that S127 in $\alpha V\beta 6$ does not directly engage the MIDAS metal ion in liganded $\alpha V\beta 6$, but do not comment on the significance of that finding. This would benefit from additional discussion.

This is a good suggestion. Previous Fig. 6 is moved up to Fig. 3, and additional sentences are added to this section.

6. At the end of the Results section, this reviewer was surprised that the interesting $\alpha V\beta 6$ mutants in Figure 5A were not discussed. These mutants are later discussed at length in the Discussion. The authors should consider reporting the $\alpha V\beta 6$ mutants data in the Results section and then focusing just on the implications in the Discussion section.

We have done so.

7. Page 6. To facilitate the interpretation of Figure 3A-F, text should be modified to "the vertical dashed lines in Fig. 3A-F mark the position of the state 1 $\alpha IIb\beta 3$ D126 carbon atom sphere."

So done.

8. Page 6. The authors should explain why movement of the S5 residue to the MIDAS should increase affinity for ligand.

So done.

9. Page 6, last line. It appears that the authors are referring to Fig. 3G rather than 2G.

Corrected.

10. Page 8. The authors note that ligand binding enhances α I helix hydrogen deuterium exchange, but in figure 4 it appears that exchange in α I helix varies over time after ligand binding, with only some time points showing enhancement.

Yes. That is not an unexpected result—only short-term exchange is affected.

11. Page 14. The authors refer to reference 18 (Hu and Luo) which showed that N8/N9 α V β 3 has similar vitronectin binding to that of WT α V β 3 whereas D8/D9 α V β 8 has decreased binding of vitronectin relative to WT α V β 8. They acknowledge the difference with the enhanced RGDLATI ligand binding they observed with mut8 (D8/D9 α V β 8), but do not provide any explanation. They allude to “vitronectin binding, rather than affinity,” but the meaning of this is not clear to this reviewer. Since Hu and Luo reported that the leg regions of α V β 8 play a role in receptor ligand binding affinity, the authors should discuss the impact of their studying the α V β 8 headpiece with an engineered disulfide bond rather than the intact α V β 8 receptor.

We have revised to “the amount of vitronectin binding to cells, rather than affinity”. The disulfide is far from the ligand binding site and so will have no effect on affinity. The headpiece will have identical affinity as the intact integrin as in the absence of a bent conformation and headpiece opening, there is no way for the headpiece to communicate with the legs. We have added sentences explaining this.

12. Page 14. The authors refer to the ADMIDAS metal ion coordinating the Asp in the β 4- α 5 loop and stabilizing its position in the outer coordinate shell of the MIDAS metal ion. The authors should consider putting the sequence of this loop in Figure 3G. What is the evidence for the interaction being in the outer shell?

We only included regions involved in conformational change and that were mutated in Fig. 3G. Octahedral Mg²⁺ coordination distances are 2.0 to 2.1Å (inner shell). Outer shell distances are considerably larger and such residues hydrogen bond to atoms in the inner shell.

13. Page 16. The description of the Fab that permits the β I to be open when the hybrid domain is either closed or open should be described in more detail.

We have added the detail that it is activating but there is little more to be said since the Fab binding site is only known from negative stain EM.

REVIEWERS' COMMENTS:

Reviewer #1 (Remarks to the Author):

The revised manuscript addresses and clarifies most points from the reviewers. Thank you for your patience and taking the time to answer these.

With regards to point #4, this was referring to lines 127-128 (line numbering helps a lot, thank you), i.e. where the authors compare regions of the molecule of the same structure, but nevertheless, their clarification in the text quantifies the statement, thank you

Point #5, hopefully (and not impossible) I am the only one who finds the three ways one residue is described confusing

Point #15, while it is trivial to convert molarity into weight/volume with the knowledge of the molecular weight, can you please stick to one or the other so it is easier to compare if one is a concentrated or diluted form of the sample

Point #21, ok so I appreciate that this would require to redo the experiment which would confirm the identity of the bands but given the structure and class averages perhaps not absolutely necessary (please do at least the SDS-PAGE of fractions in future manuscripts) but adding the calculated and apparent molecular weights should not be any trouble at all

Point #23, please add a sentence in the text the potential limitations of the interpretation and if available, explanation of not getting better data and for non-structure scientist, a layman's statement that the PDB validation reports are below average quality for their respective resolutions while their conclusions are sound and consistent with the evidence

Reviewer #2 (Remarks to the Author):

The authors have adequately addressed all my comments etc. I fully recommend publication of this interesting work.

Response to Referees

REVIEWERS' COMMENTS:

Reviewer #1 (Remarks to the Author)

The revised manuscript addresses and clarifies most points from the reviewers. Thank you for your patience and taking the time to answer these.

With regards to point #4, this was referring to lines 127-128 (line numbering helps a lot, thank you), i.e. where the authors compare regions of the molecule of the same structure, but nevertheless, their clarification in the text quantifies the statement, thank you.

Point #5, hopefully (and not impossible) I am the only one who finds the three ways one residue is described confusing

This is the first time I have been told it is confusing to use for amino acids the one-letter code in the figure and the three-letter code in the text. One does the former to make the figure less busy and the latter to make it easier for generalists. I have done this for many years. If generalists don't know the one-letter code, they can learn it by following the text and correlating with the figures.

Unfortunately, it is also necessary to use the DXSXS type of notation also, to allow comparisons between equivalent residues in the, b3, b6, and b8 structures, all of which are shown in figures.

Point #15, while it is trivial to convert molarity into weight/volume with the knowledge of the molecular weight, can you please stick to one or the other so it is easier to compare if one is a concentrated or diluted form of the sample

We start with a molar excess of one component over the other, which is important to show the reader what to expect in the gel filtration pattern- the component in excess will elute later in gel filtration than the complex. So we use molar units. Protein always gets diluted during gel filtration, so the molar concentration will be lower, because of dilution. However, the relative molar ratios will change during gel filtration depending on the peak we collect. We say "(~5 µg/ml, as estimated by A_{280})". We use the "~" symbol meaning approximately because it is the estimate directly from the absorbance measured from the spectrophotometer in line between the gel filtration column and the fraction collector. We apply fractions directly to grids. For grid preparation, we never think in terms of moles, and actually µg/ml is more appropriate because we are covering a grid and larger proteins will take up more room.

Point #21, ok so I appreciate that this would require to redo the experiment which would confirm the identity of the bands but given the structure and class averages perhaps not absolutely necessary (please do at least the SDS-PAGE of fractions in future manuscripts) but adding the calculated and apparent molecular weights should not be any trouble at all.

It would be some trouble, and more important, it would not be accurate. Gel filtration does not give molecular weight, only a diffusion coefficient. Converting to molecular weight requires the assumption that the proteins are globular. As shown in the EM, the complexes are highly irregular in shape, and not globular.

Point #23, please add a sentence in the text the potential limitations of the interpretation and if available, explanation of not getting better data and for non-structure scientist, a layman's statement that the PDB validation reports are below average quality for their respective resolutions while their conclusions are sound and consistent with the evidence

The potential limitations are clearly evident in all the objective data in the PDB file itself and in the validation report which appears on-line when one visits the PDB to download the coordinate and structure factor files.

The explanation for below average density for the resolution is that standards changed after Karplus and Diederich found an objective method (CC1/2) for setting the resolution threshold. This results in the use of data to about 0.2 or 0.3Å higher resolution than previously used. Crystallographers used to throw away a lot of good data. Because of this change in practice, all of the structures my lab deposits look somewhat "below average quality."

Beyond that, there is almost never a good explanation for why data quality is better or worse in particular crystals or for particular proteins, and there are too many possible explanations to go into. We just have to try many conditions, including dehydration which worked for us, and shoot with X-rays hundreds of crystals to find the best.

The validation report is for the quality of the density relative to resolution and is just one of many possible ways of evaluating quality. Another one is the quality of the model we build, which is what all biologists will need the most. And thus readers should pay particular attention to the MolProbity percentiles in Table 1 that show that our structural models (the *.pdb coordinate files we deposit) are far better than average for their resolution, indeed in the 98-100th percentiles. These results show that our data are actually of very good quality because we can refine against that data to obtain structures that are extremely high in quality for resolution!

Reviewer #2 (Remarks to the Author):

The authors have adequately addressed all my comments etc. I fully recommend publication of this interesting work.

Thanks so much. That concludes my comments.